# Tissue fluidity mediated by adherens junction dynamics promotes planar cell polarity-driven ommatidial rotation

Nabila Founounou[1,5], Reza Farhadifar [2,3,5], Giovanna M. Collu [1,5], Ursula Weber[1], Michael J. Shelley [2,4] & Marek Mlodzik [1✉]

The phenomenon of tissue fluidity—cells' ability to rearrange relative to each other in confluent tissues—has been linked to several morphogenetic processes and diseases, yet few molecular regulators of tissue fluidity are known. Ommatidial rotation (OR), directed by planar cell polarity signaling, occurs during *Drosophila* eye morphogenesis and shares many features with polarized cellular migration in vertebrates. We utilize in vivo live imaging analysis tools to quantify dynamic cellular morphologies during OR, revealing that OR is driven autonomously by ommatidial cell clusters rotating in successive pulses within a permissive substrate. Through analysis of a rotation-specific *nemo* mutant, we demonstrate that precise regulation of junctional E-cadherin levels is critical for modulating the mechanical properties of the tissue to allow rotation to progress. Our study defines Nemo as a molecular tool to induce a transition from solid-like tissues to more viscoelastic tissues broadening our molecular understanding of tissue fluidity.

[1] Dept. of Cell, Developmental, & Regenerative Biology, Graduate School of Biomedical Sciences, Icahn School of Medicine at Mount Sinai, One Gustave L Levy Place, New York, NY 10029, USA. [2] Center for Computational Biology, Flatiron Institute, Simons Foundation, 162 5th Ave, New York, NY 10010, USA. [3] Department of Molecular and Cellular Biology, Harvard University, 52 Oxford St, Cambridge, MA 02138, USA. [4] Courant Institute, New York University, 251 Mercer St, New York, NY 10012, USA. [5] These authors contributed equally: Nabila Founounou, Reza Farhadifar, Giovanna M. Collu. ✉email: marek.mlodzik@mssm.edu

During morphogenesis, complex tissue patterns and structures are formed from simple sheets of cells. Cell shape change and positional rearrangements facilitate these morphogenetic processes and yet overall tissue integrity has to be maintained throughout[1–3]. Cells must adhere to their neighbors to maintain the integrity and also to transmit mechanical forces throughout the developing tissue[1,2,4,5]. Adherens junctions (AJs) are the critical site of cell adhesion and their linkage to the cytoskeleton allows cells to both respond to and generate tension at the tissue level. E-cadherin (E-cad) is a core component of AJs: through its extracellular domain it binds E-cad on neighboring cells and its intracellular domain provides an anchor point for the actin cytoskeleton through binding partners such as the catenins and vinculin[6,7]. E-cad molecules form clusters of varying sizes that are dependent upon both *trans* and *cis* interactions with other E-cad molecules[8,9]. AJs are dynamic structures and their regulated turnover mediates tissue remodeling. Depending on the size of E-cad clusters and their recycling rate to/from the cell surface, AJs can be stable complexes that withstand tension, or dynamic complexes that allow a remodeling of cell contacts to relax tension at the tissue level[5,8,10].

Tissue fluidity in epithelia has been defined as the ability of cells in a confluent tissue to exchange neighbors and yet maintain intercellular junctions to ensure epithelial integrity and associated barrier functions[3,11,12]. Throughout morphogenetic processes, tissues need sufficient fluidity to allow cell intercalation and tissue shaping: tissues that are solid-like resist the applied forces whereas more fluid-like tissues can respond to stresses by junctional remodeling and allowing processes such as tissue flow or shape change. Tissues are examples of active materials, where local interactions between the basic elements, in this case cells, consume energy and govern the large-scale behavior. Cell adhesion through AJs is a major mediator of such local interactions. Convergent extension is one example of how tissue fluidity drives axis elongation in both vertebrates and invertebrates[2,4,5,10,13]. Cells undergo coordinated shape adjustment and movement with respect to their neighbors to change the dimensions of the tissue and elongate the anterior−posterior body axis. Although progress has been made at examining the biophysical dynamics of unidirectional movements of epithelia, such as convergent extension in the *Drosophila* embryo or tissue flow in the *Drosophila* thorax and wing[2–5,10,11,13–15], the molecular signals regulating the degree of fluidity at the tissue level remain largely unknown, particularly during tightly-localized and temporally- discrete morphogenetic events.

The *Drosophila* eye is uniquely suited as a model to address these questions. The eye is comprised of ~700 identical individual units, termed ommatidia, which sequentially undergo morphogenetic processes, including ommatidial rotation[16,17]. Each ommatidium contains a cluster of eight photoreceptors (R1–R8 cells) and additional accessory cells, which are progressively recruited to each cluster following the anterior progression of the morphogenetic furrow (MF) through the eye imaginal disc (Supplementary Fig. 1a–c)[16]. As the furrow progresses across the tissue, it induces uncommitted cells to differentiate and recruit their neighbors in a stereotypical sequence to initially form 5-cell pre-clusters, which are separated from surrounding pre-clusters by 2–4 layers of neighboring interommatidial cells (ICs) (Fig. 1a and Supplementary Fig. 1a–c). Planar cell polarity (PCP)-signaling induces the differential specification of R3 and R4, which breaks the symmetry of the pre-cluster and is followed by the rotation process with the direction of rotation being governed by the R3/R4 positioning (in the R3 to R4 direction), clockwise dorsally or counterclockwise ventrally[16]. This creates a line of mirror-symmetry along the D/V midline, the equator (Fig. 1a and Supplementary Fig. 1a–c). The rotation process is needed for

proper axonal bundling of R-cells to allow their correct connectivity in the neurons in the optic lobes[18,19], and ceases after each ommatidium rotates 90°.

All cells in the developing eye are connected via E-cad-containing AJs, although the ommatidial (pre)cluster cells upregulate E-cad relative to the ICs to maintain cluster cohesion[20]. In addition, other adhesion systems play a role in the eye with N-cadherin and the atypical cadherin Flamingo also being upregulated in the ommatidial cluster[21,22], whereas the cell-adhesion molecules Echinoid and Friend of Echinoid have more dynamic expression patterns throughout the eye during the rotation process[23]. In addition to cell−cell adhesion, cell-extracellular matrix (ECM) interactions are also required for eye morphogenesis[24]. Photoreceptor cells express the *Drosophila* homolog of β1 Integrin, Myospheroid, which localizes to the outside of the R-cells and forms a cup-like shape around each cluster, where it interacts with the ECM[24]. Mutations in any of these components lead to rotation defects, with clusters either rotating asynchronously, or too little or too much (reviewed in[17], also refs. [20–24]).

The genetic basis of cell specification and the signaling pathways governing ommatidial rotation have been well studied (reviewed in refs. [17,25], and also e.g., refs. [26–30]), however, the cell biological and biophysical aspects of rotation remain poorly understood. Indeed, due to the lack of specific genetic tools to manipulate signaling in the ICs, the vast majority of past studies have focused on rotation cues emanating from the ommatidial cluster itself, rather than its surrounding environment. From a biophysical standpoint, it remains unclear how the rotating groups of cells coordinate their collective movement within a field of, what has been considered to be, immobile epithelial ICs[17,31]. Also, whether the forces driving rotation comes from the pre-cluster itself (autonomous) or from the surrounding ICs (nonautonomous) remains unresolved. Moreover, biophysical mechanisms controlling complex cellular behavior such as chiral rotation of groups of cells in a field of non-rotating confluent cells, and the effect of tissue fluidity on the dynamics of such movements have not been addressed. We show here that rotation requires a specific level of tissue fluidity to enable the precise progression and cessation of rotation. Tissue fluidity is controlled by Nemo (Nmo, a conserved MAPK family member)[31–34], which we show regulates junctional E-cad dynamics, and thus overall tissue fluidity and mechanical properties.

## Results

**An in vivo live imaging approach to analyze ommatidial rotation.** The primary method of analyzing and quantifying rotation has been to determine ommatidial orientation within the successive rows of pre-clusters posterior to the furrow in fixed tissue (Supplementary Fig. 1a–d). This identified the genetic basis of rotation and the importance of PCP factors in establishing rotation direction[35,36]. However, a mechanistic understanding of ommatidial rotation, and PCP-directed movement more broadly, requires spatiotemporal measurements in real-time of cellular processes and associated behaviors, including force generation and transduction mechanisms[14]. We established a live imaging protocol of pupal eye tissue in vivo during rotation, focusing on E-cad to monitor cellular junctions as a readout of cell behavior, and visualized the process with an E-cadherin::GFP knock-in *Drosophila* line (see "Methods"). We observed active furrow progression generating new rows of ommatidial pre-clusters and their subsequent maturation and rotation (Supplementary Fig. 1e–i; Supplementary Movie 1). To investigate the mechanistic basis of the rotation process, we developed image processing software to quantify the temporal variation of various force-

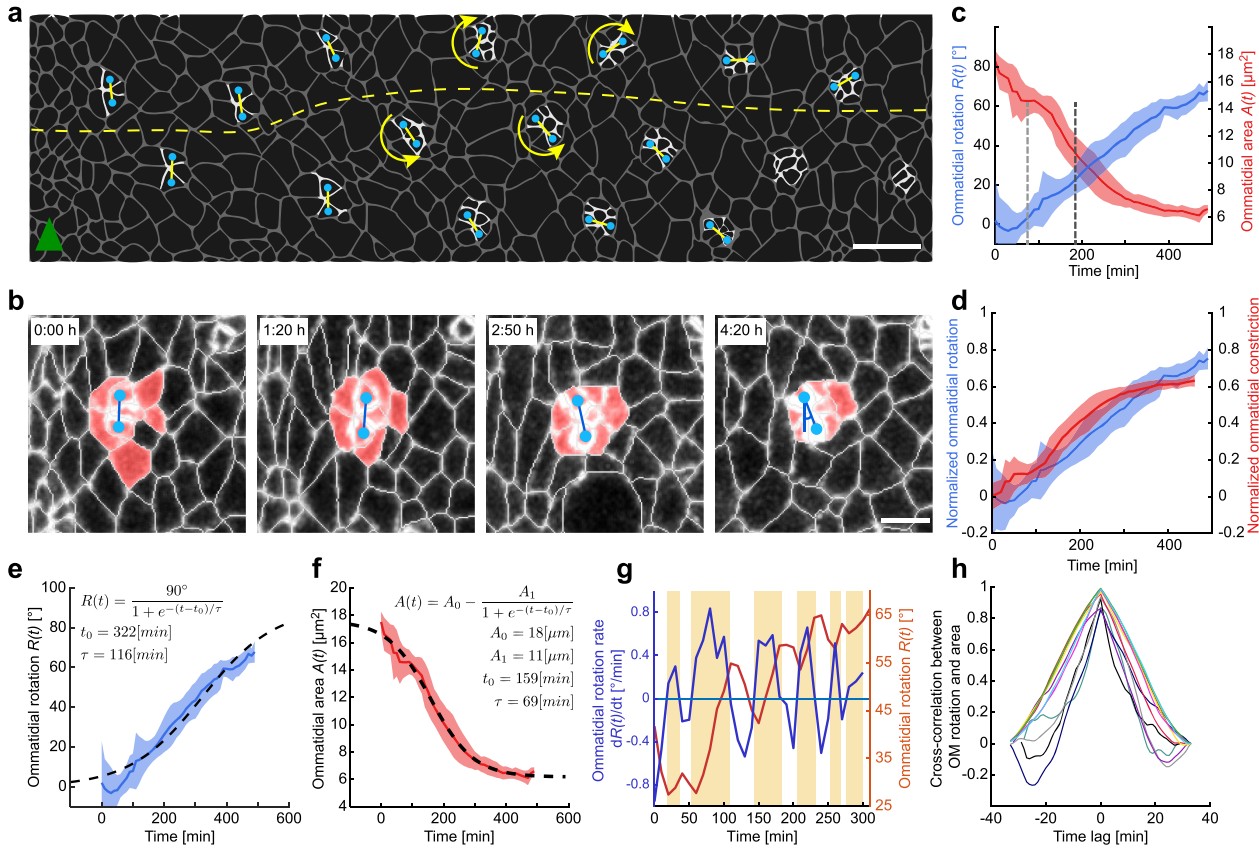

**Fig. 1 Dynamics of ommatidial cluster rotation reveal a pulsatile, continuous process. a** Representation of ommatidial clusters posterior to the morphogenetic furrow (MF, indicated by green arrowhead) in the region of the equator (yellow dashed line); blue dots denote centroids of photoreceptors R2 and R5, highlighting the angle of clusters relative to the MF and thus rotation of clusters from their initial position as they mature (away from the MF) (Supplementary Fig. 1a–c for more details). **b** Snapshots at the indicated timepoints from a movie of E-cad::GFP depicting ommatidial pre-cluster maturation and rotation. Image segmentation is shown in the white overlay. See Supplementary Fig. 1 for more details. Centroids of R2 and R5 cells were used to measure the degree of rotation over time as shown in the right panel. Cells highlighted in red are the eight photoreceptors of the cluster. **c** Rotation (blue) and apical surface area (red) of ommatidial clusters as a function of time. Measurements from different ommatidia were temporally aligned with the time when R7-cell joined the cluster (highlighted in this and subsequent graphs by dark gray dashed line, light gray dashed line highlights time point when R1/R6 join the cluster). Line shows mean, and shaded region indicates standard deviation. **d** Normalized ommatidial rotation ($Rt(90)$, blue) and normalized apical surface constriction ($1 - A(t)/A_0$, red) of individual clusters are shown as a function of time. Line shows mean, and shaded region indicates standard deviation. Note the close association between rotation and constriction. **e** Sigmoid fit (dashed line) to ommatidial rotation, $R(t)$ (line shows mean, and shaded region indicates standard deviation). The angular velocity during rotation is calculated from the fit parameters as $\Omega = 60 \times 90/4\tau \sim 11\,[°/h]$. **f** Sigmoid fit (dashed line) to the ommatidial area, $A(t)$ (line shows mean, and shaded region indicates standard deviation). The rate of ommatidial constriction is calculated from the fit parameters as $C = 60 \times A_1/4\tau \sim 2.5\,[\mu m^2/h]$. **g** Change in degree of rotation over time of an individual cluster showing $dR(t)/dt$, in blue (left axis), and $R(t)$, orange (right axis). Pulsatile behavior is marked by orange shaded boxes denoting periods, or pulses, of positive rotation with anti-rotation behavior in between such periods (white areas). **h** Cross-correlation between standardized ommatidial rotation and constriction as a function of time lag for individual ommatidia, denoted by different colored lines. Note that the maximum cross-correlation for all ommatidia occurs at zero time lag, implying a simultaneous occurrence of these two processes and a close association between them. Measurements in **c–f** and **h** are averaged over 11 clusters from two pupae in >300 min movies. OM; ommatidia. All scale bars: 3 μm.

generating mechanisms, including cell division, delamination (apoptosis), neighbor exchange, and cell movement in wild-type (*wt*) control animals, as well as quantifying rotation in genetic backgrounds in which force-generating mechanisms were perturbed.

**Rotation is a continuous, pulsatile process**. Rotation could be driven by forces intrinsic to the cluster and/or input derived from the surrounding ICs. We examined the spatial and temporal variation of cellular dynamics in *wt* eyes to determine whether any occur concomitantly with rotation and might therefore drive the process. Ommatidial rotation was measured over time, $R(t)$, as

the change in the angle between the line connecting the R2/R5 centroids and the furrow (Fig. 1a–e). Upon the formation of ommatidial 5-cell pre-clusters (row 1), the R2/R5 pair was parallel to the furrow ($R(0){\sim}0°$), and with time this angle increased (Fig. 1a–e). To obtain measurements of rotation for an extended time period, we tracked ommatidia from rows 1–4, allowing us to quantify rotation—and other cellular processes—for a period of up to 500 min (~8 h) (Supplementary Fig. 1g–i). Ommatidial clusters rotated with an average angular velocity of $\Omega = 11°/h$, fitting a sigmoid curve (Fig. 1e). This continuous progression is in contrast to the proposed two phases of 45° rotations suggested in a previous study performed in fixed tissue[32]. Moreover, the

dynamics of rotation appear pulsatile in each individual cluster within the continuum of the process (Fig. 1g), immediately demonstrating insight from live imaging.

Concomitant with rotation, the apical surface area of ommatidial clusters constricted (see pseudo-colored apical surface in (Fig. 1b). The dynamics of constriction followed a similar profile to the rotation rate, with an average constriction of ~60% of the original size with a rate $C = 2.5 \ \mu m^2/h$ (Fig. 1c–f). A cross-correlation analysis of rotation and apical constriction indicates that these two dynamic, pulsatile processes are strongly correlated without a time lag ($r = 0.98$, $p = 1.58e{-}30$, Fig. 1g, h), suggesting that the force-generating mechanism driving both processes is shared, or that one directs the other. In an effort to examine the involvement of other force generation mechanisms within the cluster, we turned our attention to the migration of photoreceptor nuclei to the apical surface. This movement is initiated once cells have been recruited to the ommatidial pre-cluster[37]. Nuclear migration can be a force generator for cell motility in mammalian cells[38]. We used the klarsicht mutant ($klar^{\Delta 1-18}$), which affects the apical positioning of pre-cluster cell nuclei, and exhibits randomized nuclear positioning along the apico-basal axis[39,40]. Rotation proceeded in klar mutants as in wt (Supplementary Fig. 2), demonstrating that nuclear migration is not a major force generation mechanism during the rotation process.

**Interommatidial cells appear to serve a permissive role during rotation progression.** The rotation process requires both a force-generating feature, which appears to be autonomous to the cluster based on the above correlation with cluster constriction, and yet could also be directed by external forces from the immediate environment (e.g., via shear forces or cell adhesion), which would be a non-autonomous feature[20–24,31,34]. Other non-autonomous aspects regulating rotation could include the cellular dynamics of the eye disc as a whole. Previous studies in fixed tissues have shown that ICs near the furrow divide (second mitotic wave) and delaminate[41,42]. Further from the furrow, cell division and delamination cease[41,42] (the photoreceptors themselves are post-mitotic and do not delaminate). The coordinated waves of division and delamination appear to coincide with the onset and progression of rotation. Cell division, in particular oriented cell division, and delamination can be force-generating mechanisms that drive tissue morphogenesis[43,44], and the behavior of ICs could thus directly impact the rotation of ommatidial clusters.

We, therefore, sought to analyze the temporal change in division and delamination of ICs directly adjacent to ommatidial clusters during rotation (Fig. 2a, b and Supplementary Movie 2). Our analyses revealed that cell division peaked upon the formation of pre-clusters close to the furrow, and the fraction of dividing cells decreased gradually and ceased after $t{\sim}250$ min (Fig. 2b). Upon a decline in cell division, cell delamination peaked at $t{\sim}350$ min, and subsequently ceased at $t{\sim}450$ min (Fig. 2b). However, as the speed of rotation was consistent throughout both peaks of cell division and delamination, and rotation continued even after the decline of both, neither process appeared to directly influence rotation rate. Consistently with these data, adult eyes displayed a wild-type phenotype when cell division orientation was perturbed in the mud loss of function backgrounds (Supplementary Fig. 3a, d). It was also shown previously that the second mitotic wave can be uncoupled from patterning in the eye[45]. Similarly, inhibition of cell delamination did not affect the morphology of the adult eye (Supplementary Fig. 3a–c), suggesting that these processes are not essential for the force generation that drives rotation and its progression.

Next, we asked whether ICs could be driving rotation by shear forces arising from the directed or patterned neighbor exchange

and IC movement in the immediate environment. The movement of ICs has not been studied, and generally, ICs have been presumed to be immobile. To this end, we analyzed the neighbor exchange and trajectories of ICs that are in contact with the cluster (see "Methods") to determine whether there was a relationship between rotation rate and the rate of junction remodeling, or a coherent flow of ICs relative to the direction of rotation. We measured the temporal variation in neighbor exchange, $N(t)$, of ICs adjacent to ommatidial clusters (Fig. 2c, d, and Supplementary Movie 2): cell neighbor exchange continuously declined during rotation from 7 events/ommatidium at the formation of the pre-clusters to 4 events/ommatidium by the end of our imaging period (Fig. 2d), while the rotation of the cluster proceeded with the same angular velocity (Fig. 1c–e; the initially higher levels of neighbor exchange seem to be related to the second mitotic wave[43,44] and not to rotation progression). We can conclude that neighbor exchange is occurring throughout the rotation and that it declines as rotation nears completion.

During this period, we also measured the motion of ICs by tracking their centroids (Fig. 2e) and plotting the average displacement field of all analyzed ommatidia during a 3 h time window (Fig. 2f). If ICs were pulling the pre-cluster during rotation, we would expect to observe a coherent circular motion of ICs around the pre-cluster. Instead, these data revealed that ICs did not display such movement relative to ommatidial clusters, arguing against a role for ICs in driving the rotation process and instructing the direction of rotation (Fig. 2f). As such, taken together the data derived from our movies suggest that the clusters drive rotation autonomously (also Discussion) and that the surrounding ICs may serve a permissive role during the rotation process.

**Nmo modulates tissue dynamics.** To gain further insight into the mechanics of rotation, we wanted to examine these same parameters under conditions when rotation itself is specifically affected. With the wt data set serving as a baseline, we performed a comparative analysis of the rotation modulator nmo, a conserved member of the MAP kinase superfamily[32,33]. Although many genes are known to affect rotation, several, such as Egfr, also affect other aspects of photoreceptor fate specification[17,26,28]. We, therefore, wanted to analyze the function of a factor that largely only affects the rotation step, and use these phenotypes to understand which of the force generation mechanisms we identified in the wt eyes, are contributing to the nmo mutant phenotype. The nmo loss-of-function (LOF or $nmo^{mut}$) mutants show under-rotation of clusters, with rotation angle correlating with the amount of Nmo function[33]; the Nmo gain-of-function (GOF) phenotype, in contrast, shows a large variability in rotation angles at the population level, with both under- and over-rotation of ommatidia detectable (Supplementary Fig. 4a–d, nmo mutant and eye-specific Nmo overexpression are shown; also "Methods" and ref. [33]). Comparing Nmo LOF and GOF with wt should therefore allow us to identify cellular processes underlying the difference in rotation phenotypes.

We first measured $R(t)$ in each genotype and confirmed the rotation defects observed in fixed tissue (Fig. 3a, b). Given the correlation between constriction and rotation in wt (Fig. 1d, g, h), we examined constriction in the nmo backgrounds to determine whether changes in constriction rate could explain the rotation defects (Fig. 3a, c and Supplementary Movies 3 and 4). Comparing constriction to wt, we observed that ultimately the clusters constricted equally successfully in all three genotypes, reaching a comparable surface area (~8 $\mu m^2$)(Fig. 3a, c). However, the dynamics of constrictions were rather different

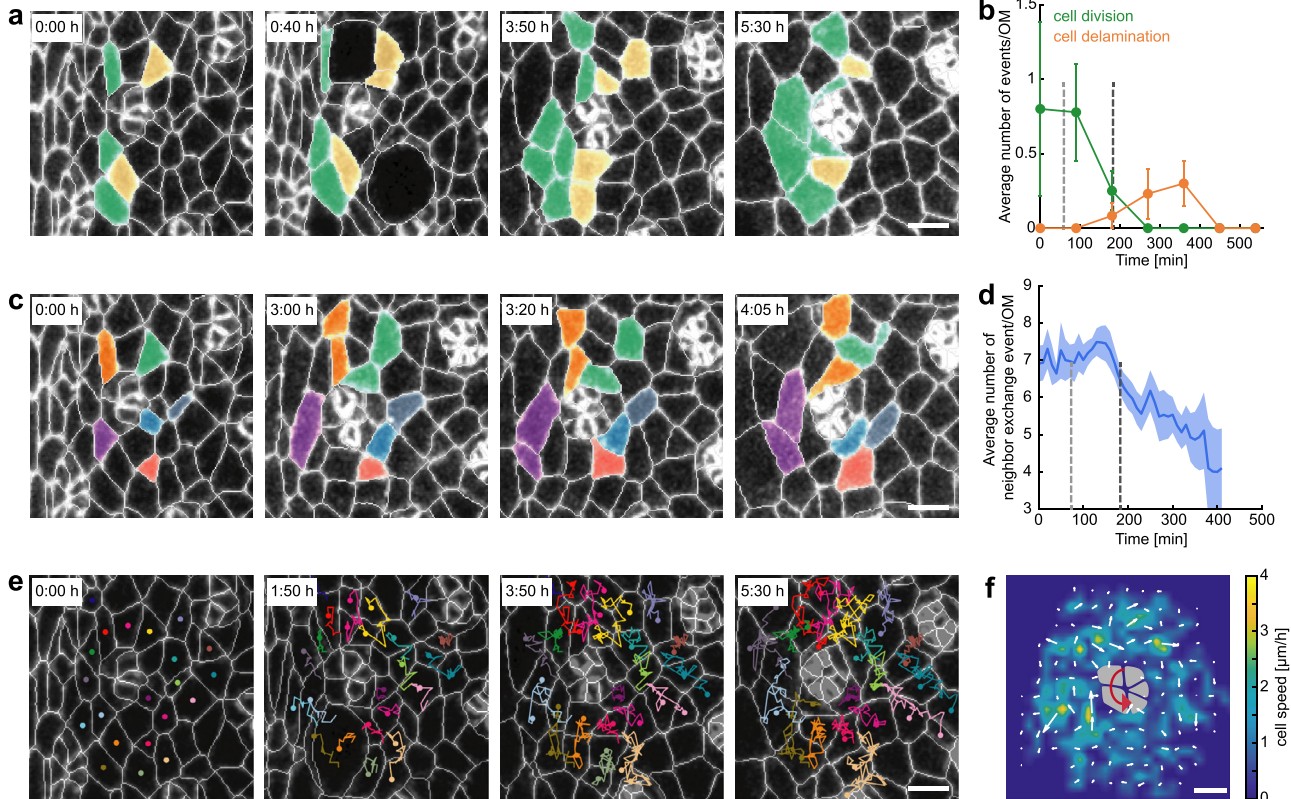

**Fig. 2 Interommatidial cells display dynamic behaviors during ommatidial rotation. a** Snapshots of the same cluster shown in Fig. 1 highlighting interommatidial cells (ICs) that divide (green) and delaminate (yellow; these cells can also divide, but at least one of the daughters delaminates subsequently). The ICs analyzed were first level neighbors of the cluster at the R7 recruitment stage. **b** Average number of dividing (green) and delaminating (orange) ICs that were the first-level neighbors of the photoreceptor cluster, as a function of time. Data were binned into 90 min time periods and were averaged over all ommatidia. The mean and standard error of the mean is shown. Gray dashed lines indicate the recruitment of the R1/R6 pair and R7, respectively (also in **d**). **c** Selected ICs, highlighted with random colors, illustrating junction remodeling and neighbor exchange. **d** Average number of ICs that were the first-level neighbors of the photoreceptor cluster and underwent neighbor exchange in between frames plotted as a function of time. Line shows mean, and shaded region indicates standard error of the mean. **e** Snapshots with trajectories of IC centroids, with dots representing the current centroid position in the respective image. **f** Average cell speed and displacement of ICs during ommatidial rotation. The trajectories of ICs from all ommatidia were pooled, and the average displacement (white arrows) and cell speed (heat map) were measured. A 5-cell pre-cluster schematic with gray cells is indicated in the middle of the plot with a curved red arrow representing the direction of rotation. Measurements in **b**, **d**, and **f** are averaged over 11 clusters from two pupae in >300 min movies. All scale bars: 3 μm.

between the three genotypes (Fig. 3d). In the LOF *nmo^mut^* tissue, constriction was a steadier process than in *wt*, exhibiting lower amplitudes of relaxation following consecutive constriction pulses (Fig. 3d–f). In contrast, in the Nmo GOF, pre-clusters failed to stabilize successive constriction pulses and exhibited higher amplitudes of relaxation between constriction pulses (Fig. 3d–f). Taken together these data suggested that the ability of the contractile apparatus to constrict the apical surface was largely unaffected in both LOF and GOF scenarios, but that the effective stabilization of contractions at the junctional level was altered. Actomyosin contractions are required for apical constriction, with a molecular ratchet required to stabilize size decrease[46–48]. Consistent with a failure of stabilization and not actomyosin contraction, MyosinII (Zipper::GFP) intensity and localization were not affected in *nmo* LOF mutant cells (Supplementary Fig. 4f), and previous studies have shown that *nmo* and *zipper* are not bona fide genetic interactors[49]. Moreover, the rotation defects seen with a reduction in MyoII or dROK function are accompanied by a failure to constrict the apical surface of affected clusters[50], which is different in appearance to the behavior in *nmo* mutant clusters. Our *nmo* data thus reveal a new category of rotation phenotype—rotation defects without apical constriction defects in the ommatidial cluster. Consistent with

this notion, our previous experiments also failed to reveal a specific genetic interaction between *nmo* and the actomyosin-associated regulators *dROK* and *RhoA* (ref. [33]). Given that Nmo phosphorylates E-cad and β-catenin in vitro and that these phosphorylation events affect the rotation process[33] (also Supplementary Fig. 4e), these data suggest that the constriction dynamics may be affected by the ability of AJs to stabilize constriction pulses and form a new cell shape through junctional remodeling. This would have a subsequent effect on the ability of the cluster to rotate. We, therefore, conclude that Nmo does not affect the contractile apparatus during rotation and our data suggest that the effect of Nmo on rotation might be to regulate junctional stability and/or dynamic remodeling following actomyosin contraction.

**Nmo levels affect tissue jamming and fluidity.** In addition to apical constriction, neighbor exchange was the other process that proceeded concomitantly with rotation. To analyze the impact of Nmo on this parameter, we monitored IC cell displacement and neighbor exchange in the *nmo* genotypes. We observed stark differences in neighbor exchange and cell displacement between *wt*, *nmo^mut^* LOF, and Nmo GOF scenarios. The neighbor

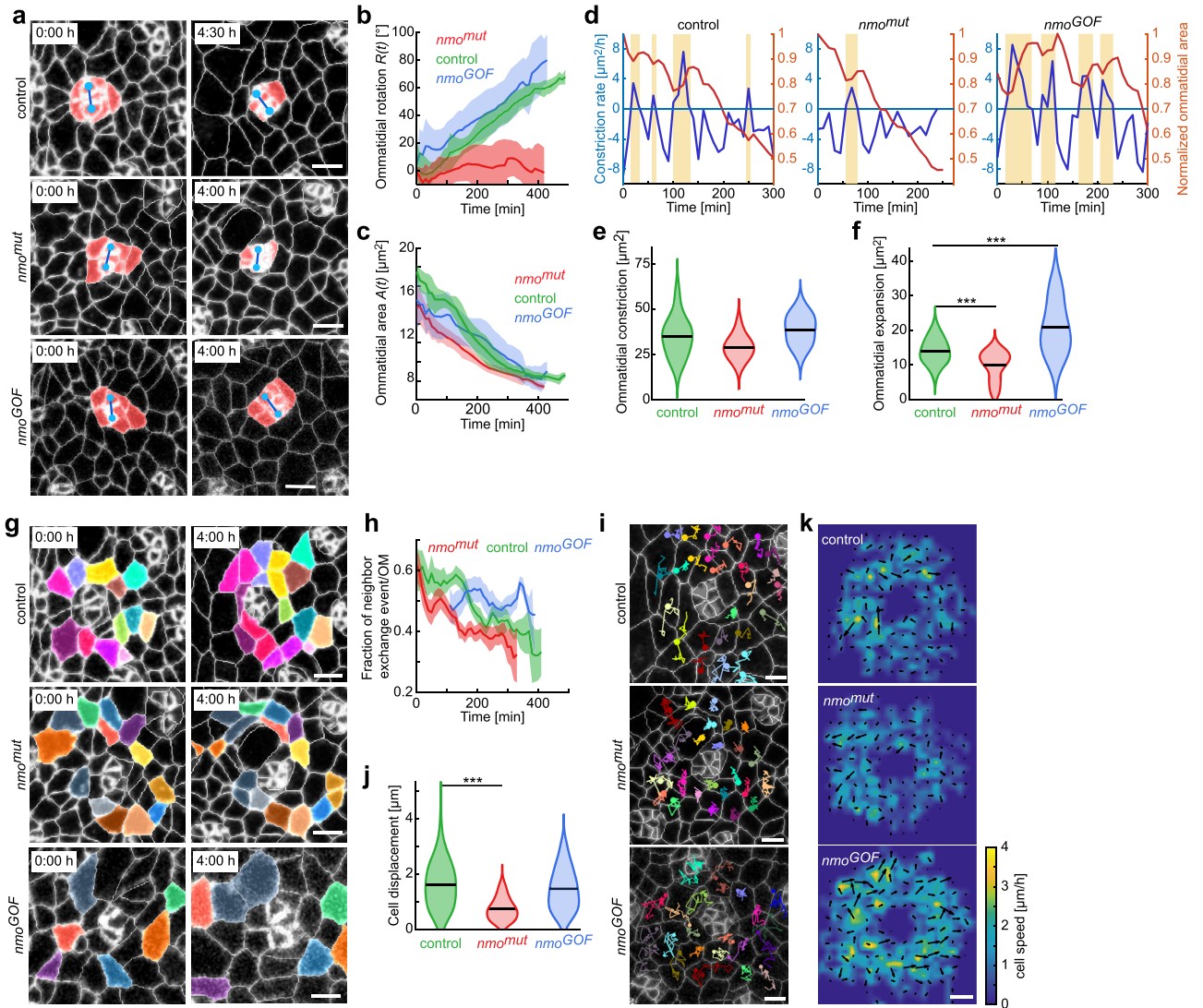

**Fig. 3 Nmo kinase affects ommatidial rotation and interommatidial cell motility. a** Snapshots of control, $nmo^{mut}$ (loss-of-function), and $nmo^{GOF}$ (gain-of-function) clusters during ommatidial rotation, respectively. The eight R-cells are highlighted in red, the R2/R5 pair is identified by blue centroid dots. **b** Ommatidial rotation as a function of time in $nmo^{mut}$ (red) and $nmo^{GOF}$ (blue), compared to the *wt* control (green). Line shows mean, and shaded region indicates standard deviation (in this panel and other equivalent panels). **c** Apical area of ommatidial clusters as a function of time for the three genotypes, as indicated. **d** Comparison of area constriction as a function of time for representative ommatidial clusters for the three genotypes. Note that all three constrict their apical area (orange line and scale on the right), but the constriction dynamics, measured by constriction rate (blue line and scale on the left), vary between the genotypes. In *wt* control, constriction happens in pulses followed by expansion (indicated by orange shaded boxes). In $nmo^{mut}$, the expansion periods were fewer as compared to control, while in $nmo^{GOF}$, expansion periods were longer and more frequent relative to control. **e** Violin plot of cluster constriction for control, $nmo^{mut}$, and $nmo^{GOF}$ as indicated, measured as the total area during constriction periods. **f** Violin plot of cluster expansion for control, $nmo^{mut}$, and $nmo^{GOF}$ genotypes measured as the total area during expansion periods. (***$p = 0.005$ and 0.004, respectively). **g** Snapshots of randomly selected ICs colored to illustrate neighbor exchange in control, $nmo^{mut}$, and $nmo^{GOF}$, respectively. **h** Fraction of ICs that were the first-level neighbors of the photoreceptor clusters and underwent neighbor exchange in between frames plotted as a function of time. **i** Snapshots with IC centroid trajectories for the three genotypes, each differently colored, in a 3 h window. **j** Violin plot of average cell displacement for ICs (***$p = 5.76 \times 10^{-32}$). **k** Average cell speed and displacement of ICs during ommatidial rotation in control, $nmo^{mut}$, and $nmo^{GOF}$, respectively. For each genotype, the trajectories of ICs from all ommatidia were pooled, and average displacement (black arrows) and cell speed (heat map) were measured. All scale bars: 3 μm. Measurements in **b**, **c**, **e**, **f**, **h**, **j**, and **k** are averaged over 17 $nmo^{mut}$ ommatidia from two pupae and 18 $nmo^{GOF}$ ommatidia from two pupae from movies >300 min. To calculate the *p*-values in (**f** and **j**), two-tailed Student's t-test was used.

exchange was markedly reduced in the *nmo* LOF mutant (Fig. 3g, h and Supplementary Movie 3) with cells maintaining the same neighbors over an extended time period, suggesting that there is more cohesion between cells, with cells appearing more adhesive to each other. In contrast, neighbor exchange was maintained at a high level in Nmo GOF tissue relative to *wt*, with cells more prone to exchanging neighbors frequently and consistently (Fig. 3g, h and Supplementary Movie 4). This cellular behavior

was confirmed by centroid trajectory plots, in which $nmo^{mut}$ LOF exhibited a mutual cell caging[12,51] (or collective trapping) phenotype with markedly reduced cell displacement of ICs as compared to *wt*, whereas Nmo GOF revealed enhanced IC movement (Fig. 3i–k; Supplementary Fig. 5h). Further analyses showed that the dynamics of ommatidial rotation and constriction were consistent within each genotype across individual pupae (Supplementary Fig. 6). Taken together, these data indicate that Nmo

GOF appeared to boost neighbor exchange (as well as the onset of cell delamination, Supplementary Fig. 5d, f), while also failing to stabilize apical constriction as discussed above. Conversely, in the LOF $nmo^{mut}$, neighbor exchange and cell delamination were both reduced (Fig. 3g, h and Supplementary Fig. 5a, c, f, g), while the stabilization of successive constriction pulses was increased (Fig. 3d, f).

Neighbor exchange and cell displacement are both key determinants of tissue fluidity, and these results suggest a gradient of fluidity from $nmo^{mut}$ (lowest) to $wt$ to Nmo GOF (highest). However, they are also impacted by other events such as cell division and delamination as seen above. To try to address tissue fluidity specifically, we turned to the geometry of the ICs. Whilst analyzing neighbor exchange, we noticed that the individual ICs had a different appearance in the $nmo^{mut}$ and Nmo GOF genotypes, as compared to $wt$; they looked more compact in the mutant and more elongated in the GOF (compare IC outlines in Fig. 4a). To quantify these phenotypes, we measured the perimeter and area of each cell (Fig. 4b, c) and both metrics indicated that $nmo^{mut}$ cells were indeed smaller and the Nmo GOF were larger than $wt$ ICs. Recent studies defined that the associated shape index[52–54], $p_0 = L/\sqrt{A}$ ($L$, cell perimeter; $A$, cell area), of cells in a given field is a way to measure not only their compactness, but also these values can be correlated with overall tissue fluidity[51–56], with the observation that the higher the average shape index value the more fluid-like a tissue behaves. In particular, a jamming transition has been proposed to occur at a value of 3.81 (ref. [53]), below which the tissue is jammed and behaves as a solid, and above which the tissue is unjammed and can flow like a liquid[51,52,55,56]. Consistent with their more compact appearance, $nmo^{mut}$ cells have a significantly lower shape index than $wt$, closer to the jamming transition point (Fig. 4d and color scheme in Fig. 4a). Conversely, Nmo GOF cells with their elongated appearance have a higher shape index, suggesting the tissue is more fluid (Fig. 4a, d). The standard deviation of the shape index is also a measure of tissue jamming[52]. The more variable the shape index within a field of cells, the more fluid the tissue. Accordingly, by this measure, the $nmo^{mut}$ cells are less variable, and Nmo GOF cells are more variable than the $wt$ control cells (Fig. 4e). Subsequent work has suggested that tissue packing also impacts the jamming transition point[55]. The pupal wing is the *Drosophila* model tissue that has been most studied in terms of cell packing and dynamics[11,13,57–59]. We thus analyzed whether $nmo$ also affected cell packing and behavior in the wing. We generated $nmo$ LOF clones or Nmo GOF patches of tissue next to $wt$ cells and analyzed cell perimeter and area, and calculated the associated shape index for each group. Consistently with data from the eye, $nmo^{mut}$ LOF cells were smaller and more tightly packed than $wt$, with a low shape index (Fig. 4f–i), and Nmo GOF cells were larger with a higher shape index than $wt$ (Fig. 4f–i). Adherens junction remodeling has been shown to directly affect cell packing and movement in the pupal wing. Hexagonal packing is dependent upon the recycling of E-cad, with defects in endocytosis changing the polygon distribution as pentagonal cell junctions are less easily remodeled to form hexagonal shape. Accordingly, we find an altered polygon distribution in $nmo^{mut}$ LOF wings (Supplementary Fig. 7e). Interestingly, this process is also dependent on PCP input[57], and p120 catenin is also known to regulate E-cad internalization and thereby affect cell shape and tissue level stress response during wing morphogenesis[11]. Strikingly, *p120* seems to have the opposite phenotype to $nmo$ LOF—*p120* mutant cells are more elongated due to higher internalization rates of E-cad and thus remodeling in response to the tissue level tension[11]. Thus, changes in Nmo levels affect not only cell movement but also cell packing and dynamics in multiple tissues undergoing morphogenesis and might thus act generally in tissue packing and associated behavior (also "Discussion"). Nmo GOF tissue appeared to be more fluid-like compared to $wt$, based on both cellular geometry and behavior (Fig. 4). Conversely, $nmo$ LOF cells behaved in a caged manner, barely moving beyond their size and adopting the morphology of locomotion-inhibited confluent cells in a solid-like tissue (Fig. 4f–i). In each case, the degree of tissue fluidity affects the ability of the cluster to rotate the full 90°, either restricting ($nmo$ LOF) or removing the regulatory input that can determine the appropriate stopping point (Nmo GOF). Consistent with the notion that $nmo^{mut}$ LOF cells are producing more tension, reflected in smaller apical areas and reduction in the most energy preferred cell shape, the hexagon[57], mutant wing cells are both smaller and display less prominent hexagonal appearance (Supplementary Fig. 7e, f; this notion is supported by previous theoretical and experimental studies arguing that higher cellular tension is linked to a smaller apical cell area and reduced junctional remodeling between cells, affecting the distribution of hexagons and pentagons[57,58]). Taken together these observations led us to examine steady-state surface E-cad levels and E-cad turnover rates in each of our genotypes.

**Nmo regulates E-cad levels at the AJs.** We used the pupal wing to analyze the level of E-cad at the junction, because of their homotypic and larger cell size, and the ability to tightly regulate the spatial pattern of Nmo overexpression, both features make it more amenable for such studies as compared to the eye. In order to compare E-cad levels accurately between the genotypes, we calculated the fold change in steady-state E-cad levels relative to neighboring $wt$ tissue, for either the $nmo^{mut}$ LOF or Nmo GOF (overexpression) genotypes. $nmo^{mut}$ LOF clones were induced as above, and for the GOF experiment we used *patched-GAL4* (driving Nmo expression in a stripe along the A/P boundary in the center of the wing). Compared to control, we saw an increase (~50%) in E-cad intensity at junctions in $nmo^{mut}$ LOF cells (Fig. 5a, c and Supplementary Fig. 7g, h). Conversely, there was a decrease (~25%) in E-cad intensity in the Nmo GOF cells (Fig. 5b, c and Supplementary Fig. 7h). These data are consistent with reduced E-cad endocytosis leading to higher levels at the junctions in $nmo^{mut}$ LOF cells, and increased endocytosis leading to lower levels of junctional E-cad in the GOF cells.

E-cad trafficking is known to play a pivotal role in junctional remodeling, cell migration, and morphogenesis[60]. We sought to directly test whether Nmo regulated E-cad dynamics at AJs using Fluorescence Recovery After Photobleaching (FRAP) experiments[61]. E-cad fluorescence recovery can be achieved either through lateral movement of E-cad molecules already present at the plasma membrane, or the arrival of newly trafficked E-cad molecules (either recycled from AJs or newly synthesized protein)[62]. To specifically address the trafficking of E-cad we bleached a 10 μm-long junction between four cells (two adjacent cell boundaries) and recorded at 20 s intervals to monitor E-cad fluorescence recovery at the junction of ICs at the center of the bleached region (Fig. 5d–f; Supplementary Fig. 7a, d and Supplementary Movies 5–7). The bleaching of entire junctions between cells ensures that only newly-trafficked E-cad contributes to recovery and not molecules diffusing within the junctional plasma membrane. LOF $nmo^{mut}$ exhibited a slower initial recovery rate compared to $wt$ controls, with a longer half recovery time (Fig. 5g–i). Conversely, Nmo GOF tissue demonstrated a faster initial recovery rate with a shorter half-recovery time (Fig. 5g–i). Of note, the recovery time here is of a similar scale to the pulses of constriction and rotation described earlier (Fig. 1g, h).

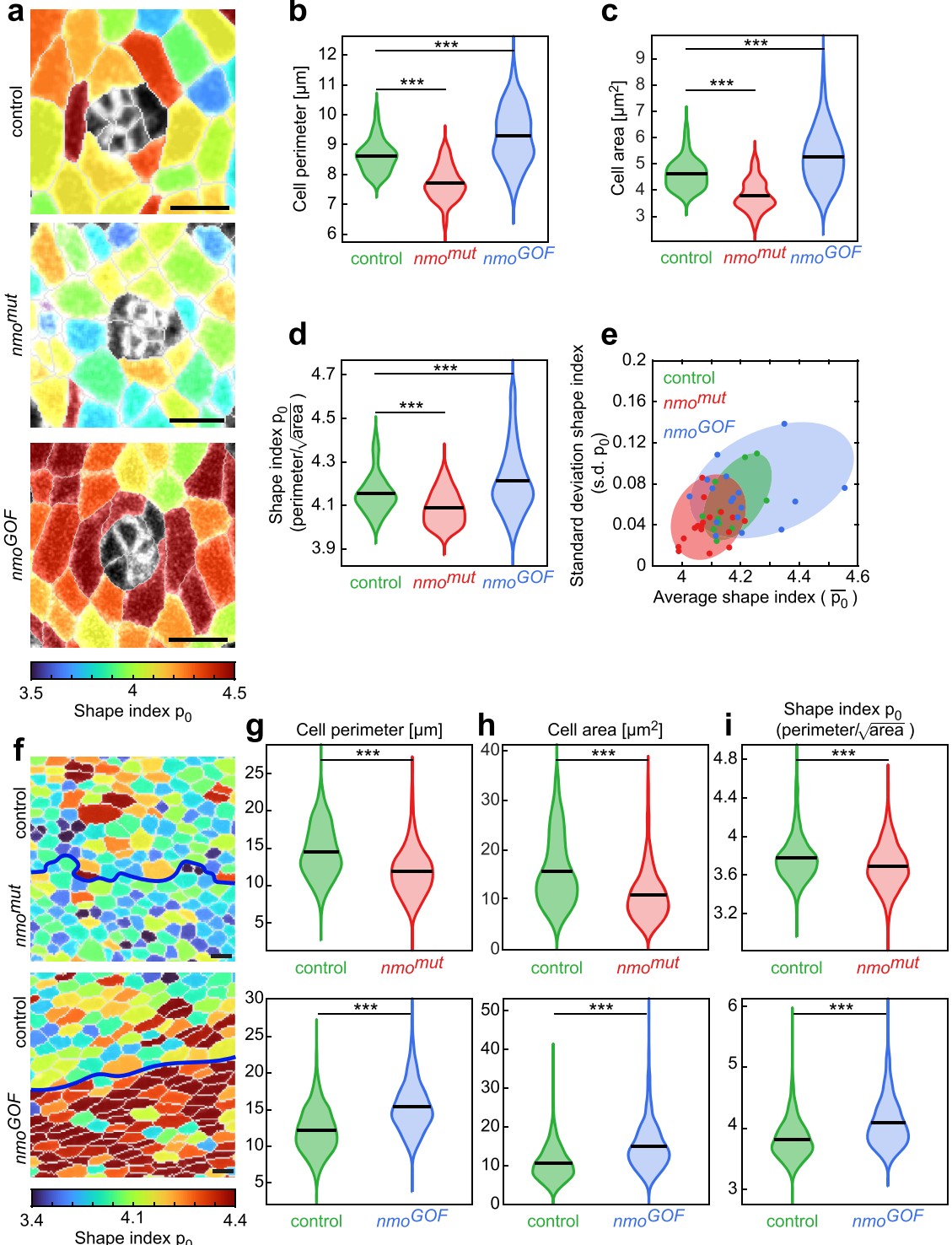

We next investigated the functional effect of altered E-cad recycling on rotation. Using the *sev > Nmo* genotype as background, we performed genetic interactions to identify factors that enhanced or suppressed the Nmo GOF variable rotation phenotype. To reduce E-cad recycling, we used a LOF allele of dynamin (known as *shibire/shi* in *Drosophila*). Removing one copy of *shi* (*shi*$^{-/+}$) was sufficient to partially suppress the NmoGOF phenotype (Fig. 5j, k), suggesting that the phenotypic effect is in part dependent upon dynamin-mediated endocytosis. We also co-overexpressed an E-cad molecule that cannot bind to p120 catenin[63], and compared it to co-overexpression of *wt*

E-cad. As p120 binding to E-cad stabilizes E-cad complexes at junctions[64], deletion of the p120 binding site should promote E-cad turnover. Indeed, co-expression of the E-cad molecule that cannot bind to p120 enhanced the *sev > Nmo* phenotype as compared to *wt* E-cad expression (Supplementary Fig. 7c), consistent with the notion that the variability in rotation angles observed with *sev > Nmo* GOF is dependent on increased E-Cad turnover. We saw that over-rotated clusters tended to center around 270°, supporting previous observations and the hypothesis that there is a stop signal in the antero-posterior axis[31]. Interestingly, Nmo GOF seemed to increase the variation around

**Fig. 4 Nmo regulates cell shape in eye and wing tissues. a** Snapshots of representative ommatidia (at stage of R7 joining clusters) in *wt* control, *nmo*[mut], and *nmo*[GOF], where interommatidial cells are colored based on their shape index ($p_0$ = cell perimeter/$\sqrt{\text{cell area}}$). The color map is shown. Scale bar: 3 μm. **b** Violin plots of the perimeter length of interommatidial cells for each respective genotype, black line shows the mean here and all subsequent panels (***$p = 2.06 \times 10^{-82}$ and $3.13 \times 10^{-27}$, respectively). **c** Violin plot of the area of interommatidial cells for the respective genotype (***$p = 1.16 \times 10^{-63}$ and $1.20 \times 10^{-21}$, respectively). **d** Violin plot of the shape index of interommatidial cells for the respective genotype (***$p = 2.44 \times 10^{-25}$ and $4.06 \times 10^{-9}$, respectively). **e** Standard deviation of shape index as a function of average shape index for all the first and second level neighbors of an ommatidium. Each ommatidium is represented as a single dot, and color-coded by respective genotype as indicated. The shaded ellipse covers the smallest area that is enclosed by the individual points (note the linear relation between these two quantities). **f** Mosaic wing tissue (fixed) at 22 h after puparium formation (APF) with cells colored based on their shape index; a *nmo*[mut] clone (*nmo*[DB] null allele) and *nmo*[GOF] cells are juxtaposed to *wt* control cells, as indicated on left. Blue line marks the boundary between experimental (*nmo*[mut] or *nmo*[GOF]) cells and control cells. The color map is shown. Scale bar: 5 μm. **g** Violin plot of the cell perimeter in control and *nmo*[mut], and in control and *nmo*[GOF] 22 h APF pupal wing tissues (***$p = 7.64 \times 10^{-62}$ and $4.49 \times 10^{-141}$, respectively). **h** Violin plot of the cell area in control and *nmo*[mut], and in control and *nmo*[GOF] wing tissues (***$p = 1.12 \times 10^{-55}$ and $4.05 \times 10^{-105}$, respectively). **i** Violin plot of the shape index in control and *nmo*[mut], and in control and *nmo*[GOF] wing tissues, respectively (***$p = 2.37 \times 10^{-14}$ and $1.35 \times 10^{-80}$ respectively). Note that the change in cell perimeter, area, and shape index between different genotypes are consistent in the eye and wing tissues. To calculate the *p*-values in (**b–d**) and (**g–i**), two-tailed Student's t-test was used. In **a–e**, the *n* values were as follows: 290 cells/11 ommatidia (control); 430 cells/17 ommatidia (*nmo*[mut]), and 450 cells/18 ommatidia (*nmo*[GOF]), each from two pupae. In **f–i**, the *n* values were as follows: >700 cells from three mosaic wings (control and *nmo*[mut]); and four mosaic wings (*nmo*[GOF]).

---

both peaks. Loss of one copy of the β1 Integrin gene, *myospheroid* (*mys*[−/+]) also suppressed the *sev > Nmo* GOF phenotype (Supplementary Fig. 7b), which is interesting because *mys* LOF also causes asynchrony/variability in rotation angle[24]—and yet overexpression of a dominant-negative Mys does not interact with Nmo LOF (ref. [33]), suggesting Mys may act downstream of Nmo.

**Nmo modulates tissue mechanics.** Loss of *p120 catenin* has been suggested to increase the rate of E-cad turnover in the wing, resulting in reduced epithelial viscosity[11]. To examine whether changes in Nmo levels, and associated E-cad turnover changes, affect viscosity in the more complex tissue of the eye, we tested the effect of *nmo* LOF and GOF on junctional tension using laser ablation experiments (Fig. 6a–f). The junctional recoil was measured after ablation of a single junction by a femtosecond laser (see Methods) and the profile of this recoil was compared between the three genotypes (Fig. 6g). These experiments revealed that LOF *nmo*[mut] exhibited higher tissue tension, as compared to *wt* control, with a greater initial recoil speed and higher final recoil amplitude (Fig. 6g, h). Furthermore, *nmo*[mut] tissue appears to be unable to absorb the resulting stress, as unsevered neighboring junctions snap due to the stress generated by the expansion of the ablated junction (Fig. 6c, red arrow; Supplementary Fig. 8, and Supplementary Movie 8). Conversely, the Nmo GOF tissue displayed the opposite phenotype with a recoil profile close to a viscoelastic tissue;[65] although the initial recoil is similar to *wt*, there is a quick resorption and contraction back to the initial distance between the two vertices followed by repair of the severed junctions in most cases (Fig. 6e–g and Supplementary Movie 9). This GOF data is consistent with increased tissue fluidity having been linked to accelerated wound healing and junction remodeling[3]. The LOF *nmo*[mut] phenotype, on the contrary, suggests that the tissue is behaving like a more solid, jammed, or pre-stressed material that is unable to absorb the additional stress generated by the severed junction and ends up yielding to the propagating tension[43,65].

## Discussion

This work defines a role for Nmo kinase in modulating tissue fluidity, which is essential for proper tissue morphogenesis. Together with previous work[33], these data revealed that Nmo regulates the dynamics of E-cad turnover at AJs directly, thus impacting junctional remodeling and tissue fluidity (see model in Fig. 6i). Our results are consistent with a model in which rotation

is powered autonomously from within the ommatidial cluster, and that the surrounding ICs serve as a permissive environment, whose fluidity regulates the degree of rotation. Actomyosin contractility is required within the cluster itself for apical constriction and rotation[50], and myosin heavy chain progressively localizes to the interface of cluster-ICs as more cells are added[49]. Intercellular adhesion is also important with adhesion molecules, including E-cad, being upregulated as cells join the cluster[20]. Consistent with the role of cell adhesion in rotation, it has been shown that clones of cells expressing the altered level of adhesion molecules can negatively impact the rotation of neighboring *wt* clusters[20,22]. Beyond the mechanical anchoring aspect of IC-cluster adhesion though, our data suggest that the fluidity of the local IC neighbors may also play a role. As the cluster rotates, its asymmetrical shape necessitates displacing ICs. The more motile the ICs, the less effort is required for the cluster to rotate. It is worth noting that basolateral Integrin-ECM mediated adhesion also affects ommatidial rotation and is linked to PCP signaling[24], and interestingly, it acts in the opposite manner to Nmo (Supplementary Fig. 7). Basolateral input also regulates the cell rearrangements studied extensively during *Drosophila* germ band extension[66]. Together these data suggest that an interplay of adhesion systems provides the right level of tissue fluidity. ICs not only insulate each ommatidial (pre)cluster from forces coming from other surrounding clusters, possibly helping to relax tension, but also serve as a permissive substrate with an ideal fluidity in *wt* to allow rotation to proceed properly. As such changes in the fluidity level of the entire eye disc tissue are critical and when this is outside the correct range it is deleterious to the process. Regulation of the IC environmental fluidity is a parsimonious way to provide robustness to the development of the eye as a whole: over 700 identical units undergo the same 90° rotation step and regulation of the IC environment is one way to ensure uniformity in morphology and functionality.

In contrast to previous studies investigating the relationship between E-cad and the actomyosin cytoskeleton showing that mechanical forces affect junctional dynamics[67–69], our data indicate that the control of junctional E-cad levels and dynamics can directly modulate tissue tension, viscoelasticity, and stress, thus affecting tissue fluidity and cell motility. Although we cannot exclude regulation of the actomyosin cytoskeleton by Nmo through its phosphorylation of E-cad/β-catenin complexes, the largely normal apical appearance of R-cells within the cluster, which is in contrast to direct effects on the actomyosin cytoskeletal regulation causing severe disruption to the R-cell clusters[50], suggests that it is indirect at best; addressing this question here is

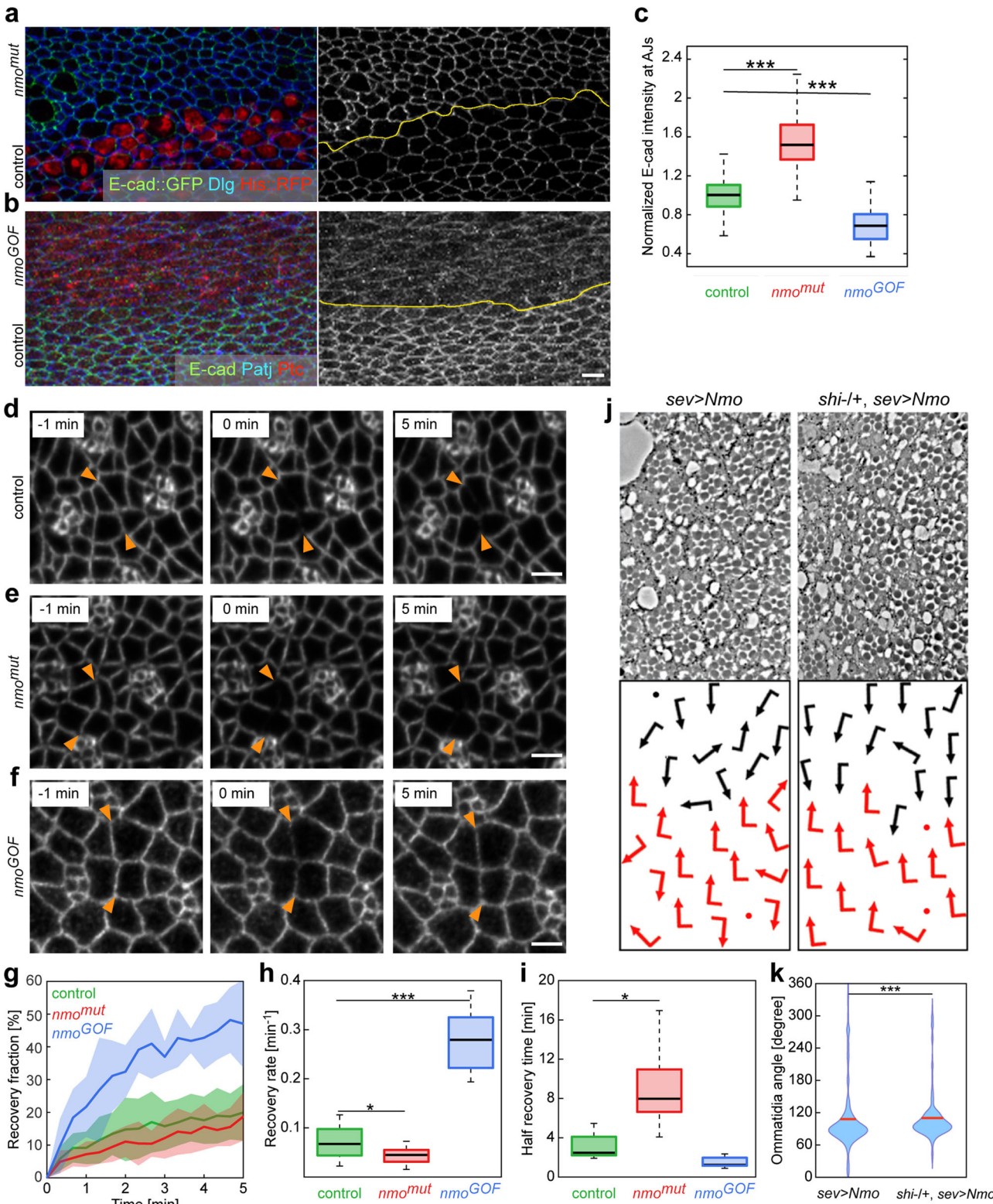

difficult due to the inherent spatial and temporal resolution requirements imposed by the tissue (small cell size, and long-timescale), which prohibit effective monitoring of the rapid cytoskeletal dynamics. Moreover, manipulating actin or myosin function directly alters the gross morphology of the clusters, precluding accurate rotation analysis of mutant phenotypes[50]. Our data support a model of linear coupling between E-cad and actomyosin that could explain the change in the mechanical

properties resulting from the modulation of E-cad levels at the junction: the more stable E-cad at the junction the more the tension generated by the cytoskeleton can be transmitted across the tissue, which results in a more solid-like, jammed tissue behavior.

Junctional dynamics have been well studied in the convergent extension model of germ band elongation in *Drosophila* embryos, as the A-P axis lengthens. Here, actomyosin force-generating

**Fig. 5 Nmo controls E-cad recycling and affects junctional E-cad steady state levels. a, b** Mosaic 22 h after puparium formation pupal wings (fixed tissue) with either *nmo*[mut] clones (**a**, *nmo*[DB] null allele) or *nmo*[GOF] (**b**, overexpressed under *ptcGal4* driver control) demarcated by the yellow line showing in (**a**): E-cad::GFP (green and monochrome), Discs large (Dlg, blue) and Histone2A::RFP (His::RFP, red, marking *wt* control cells); and antibody staining of E-cad (green, monochrome) in (**b**), Patj (blue), and Ptc (red, marking *nmo* GOF overexpression region). Note a visual increase in junctional E-cad level in *nmo*[mut] cells (**a**) and a decrease in *nmo*[GOF] cells (**b**) relative to *wt* controls (see Ext Data Fig. 7h for monochrome presentation of control Dlg and Patj staining). Scale bar: 5 μm. **c** Boxplot of normalized intensity of junctional E-cad levels averaged over 300 adherens junctions from 3 mosaic wing tissues in control, *nmo*[mut], and *nmo*[GOF] cells (mean and standard deviation; ***$p = 4.74 \times 10^{-119}$ and $1.19 \times 10^{-53}$, respectively). **d–f** Snapshots of eye tissue Fluorescence Recovery After Photobleaching (FRAP) experiments of E-cad::GFP in control, *nmo*[mut], and *nmo*[GOF]. Bleaching occurred over a 10 μm junction along two adjacent cell boundaries (shown by orange arrowheads). Scale bar: 3 μm. **g** Measured recovery fraction after photobleaching as a function of time for control, *nmo*[mut], and *nmo*[GOF]. Line shows mean, and shaded region indicates standard deviation. **h** Boxplot of initial recovery rate after photobleaching for the respective genotype measured as the slope of the fitted exponential curve at $t = 0$ (see Supplementary Information; *$p = 0.03$; ***$p = 2.77 \times 10^{-6}$). **i** Boxplot of half recovery time after photobleaching for the respective genotype measured from fitted exponential curve (see Supplementary Information; *$p = 0.01$). **j** Tangential sections of adult eyes (showing a region around the equator) of the indicated genotypes with corresponding schematics below. Chiral ommatidia are depicted as black (dorsal) or red (ventral) flagged arrows. Note that the *sev > Nmo* GOF phenotype (left panels) is suppressed by heterozygosity for dynamin (*shi*[FL54]/+, *sev > Nmo*; right panels). **k** Quantification of eye sections from genotypes shown in (**j**) presented as violin plots of ommatidial angles in adult eyes (***$p = 1.71 \times 10^{-7}$). Red line indicates mean value. Note that *shi*−/+ heterozygosity causes a reduction of widespread angle distribution and loss of additional smaller peak around 270°, as compared to *sev > Nmo* alone. In **c**, three mosaic *nmo*[mut] wings from two pupae and four *nmo*[GOF] mosaic wings from two pupae were analyzed. Box plots in panels **c**, **h**, and **i**: black line indicates the median, the edges of the box are 25 and 75% percentiles, and the whiskers extend to the most extreme datapoints. Panels **g–i** are averaged over 10 control, 9 *nmo*[mut], and 5 *nmo*[GOF] ommatidia. Panel **k** is averaged over 408 ommatidia from 3 eyes for *sev > Nmo* and 330 ommatidia from 3 eyes in *shi*−/+, *sev > Nmo*. To calculate the p-values in **c**, **h**, and **i**, two-tailed Student's t-test was used. To calculate *p*-values in **k**, two-sample Kolmogorov−Smirnov test was used.

networks undergo periodic contractility and direct intercalary movements[46,48]. Although similar to the pulsatile contraction and rotation we see in ommatidia, the timescales are different: of the order of seconds in the contractility of single cells[46,48], whereas each pulse is several minutes in the context of whole ommatidia. However, the phenotype observed in the Nmo GOF, of apical constriction followed by periods of relaxation, strikingly resembles that of Rab35 LOF during germ band extension; apical area oscillations do occur and contractile steps are functioning, but there is a reversal of movement and junctions ultimately fail to shorten[47]. Rab35-dependent ratchet-like behavior also occurs during mesoderm invagination, where Rab35 is planar polarized and dynamic, and is enriched at the interfaces of intercalating cells. Rab35 functions as an endocytic hub, and has been suggested to direct E-cad complexes for endocytosis and allow junctional remodeling. Regulation of tissue fluidity has also been shown to be important during germ band extension and mesoderm invagination (see below). Our combined studies (here and ref. [33]) on the role of Nmo kinase in phosphorylating the E-cad/β-catenin complex also closely mirror the work of Tamada and colleagues in showing that Abl kinase phosphorylates β-catenin during axis elongation in the *Drosophila* embryo[70]. More recent work has shown that convergent extension movements in *Xenopus* require local heterogeneity in junctional mechanics at the level of individual junctions[71]. Local stiffness at vertices dictates shortening and fluid-like directed motion on a sub-cellular scale. Cadherin *cis*-interactions determine local junctional stiffness and impact on vertex activity. Cadherin3 clusters undergo dynamic fluctuations in size and these temporally correlate with, and precede, pulsatile junctional shortening. It will be interesting therefore to determine how Nmo impacts E-cad clustering at the molecular level.

AJ stability and dynamics are required for junctional remodeling and cell shape changes, cell processes that are central to tissue fluidity during morphogenesis[72]. Our observations support the hypothesis that a more fluid-like tissue, i.e., permissive, would allow a higher degree of rotation due to a decrease in resistance between the cluster and ICs, whereas a more rigid tissue would decelerate, if not block, the rotation process as the forces between the cluster and the ICs cells increase. Further, our data suggest that changes in Nmo levels directly affect AJs downstream of PCP (this work, and ref. [33]), thereby controlling tissue fluidity and

acting as a catalyst for a transition from a solid-like, jammed tissue (in the absence of Nmo) to a fluid-like environment (increased Nmo levels). The contrast between the solid-like and fluid tissues explains the differences in neighbor exchange and cell caging dynamics[54,65,73], as well as onset of cell delamination (Supplementary Fig. 5): increased junctional remodeling facilitates each of these processes.

Tissue jamming has been defined as a divergence of the viscosity of a tissue with step changes in tissue fluidity, with behavior changing between amorphous solid-like and fluid-like states[65]. The phenomenon of tissue jamming has been well documented in tissue culture models of various epithelial cell types, particularly from human bronchial epithelial cells (HBECs). In culture, HBECs behave as a fluid at first and become more like an amorphous glassy solid over time due to cell adhesion and internal friction[51]. Cells become caged whereby they can hardly move beyond their size and display decreased velocity, due in part to increased E-cad at AJs. Tissue jamming can be described as a phase transition and it also occurs in reverse, which has been described as "reawakening"[74]. Overexpression of the key endocytic regulator RAB5A is sufficient to induce an unjamming transition, with increase in locomotion of cells and shape index[74]. Similarly to our results with Nmo, RAB5A overexpression does not affect actomyosin, rather it regulates the dynamin-dependent trafficking of E-cad. RAB5A has a similar effect during wound closure and zebrafish gastrulation (see also below). Using HBECs, others have shown that the unjamming transition is molecularly distinct from epithelial-mesenchymal transition, which underlies other instances of increased cell motility[56]. HBECs underwent a change in appearance from a cobblestone-like pattern of confluent cells to more elongated and variable cell shape, similar to the change in cell shape we observed between the *nmo* LOF and Nmo GOF. The dynamics and structural signatures of the unjamming transition further resemble the Nmo GOF with increased motility, cell elongation, intact junctional integrity and epithelial A/B polarity markers, and increased stochastic but cooperative motion. The link between jamming and geometry transcends systems—from HBECs to gastrulation in the *Drosophila* embryo, and from healthy homeostasis to airway disease[52,54]. Moreover, shape index and cell packing can be used as a predictor of when tissue will undergo fluidization, as shown in convergent extension and

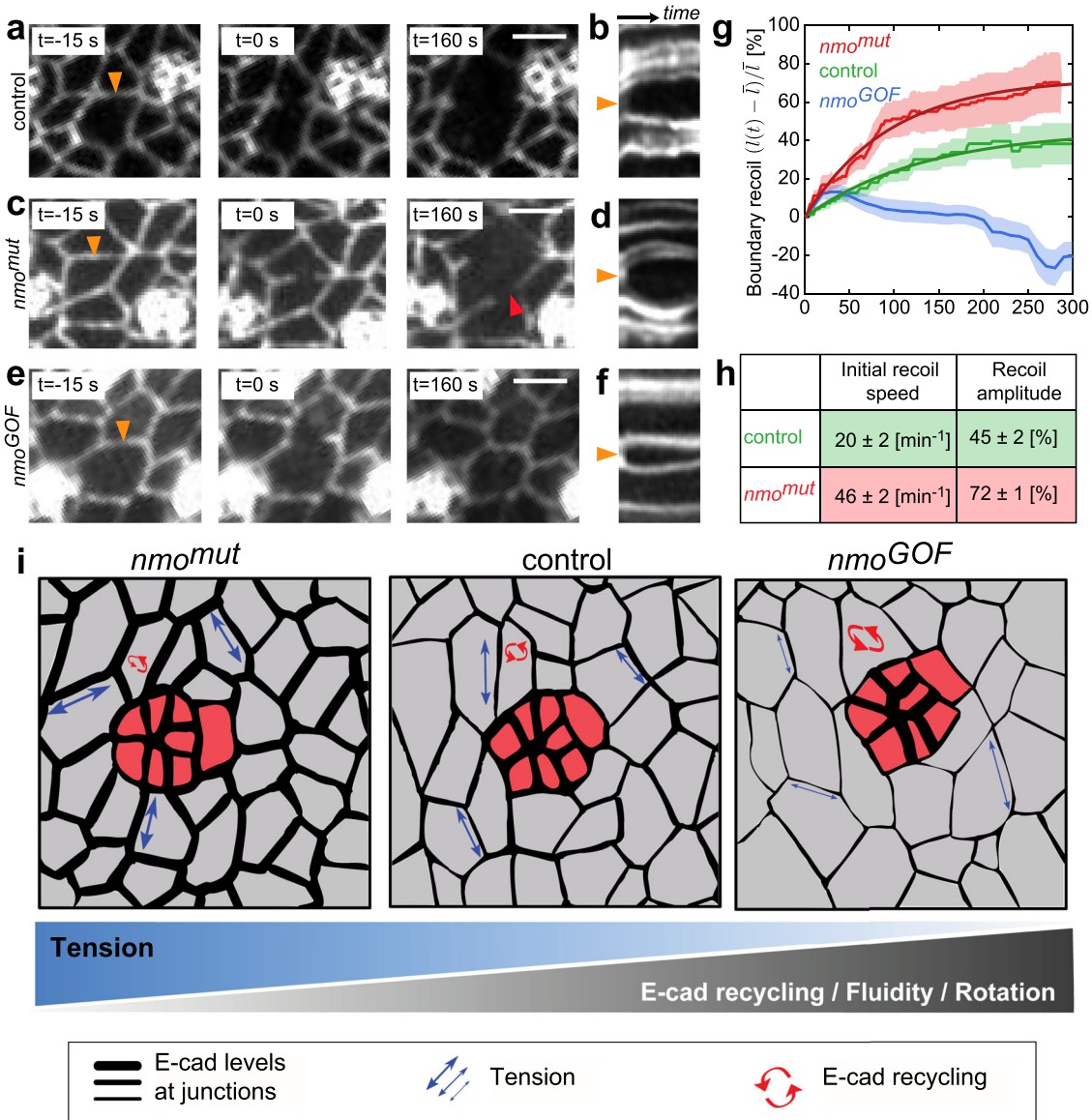

**Fig. 6 Nmo regulates tissue mechanics and tension at adherens junctions.** **a**, **c**, **e** Snapshots of laser ablation experiments performed at the level of a single junction in *wt* control, *nmo^mut^* (LOF), and *nmo^GOF^* conditions. Orange arrowheads indicate ablated junctions. Note that in *nmo^mut^* (LOF), the excessive pre-existing stress in the tissue causes ruptures in non-ablated junctions (red arrowhead in the right panel in **c**) Scale bar: 3 μm. **b**, **d**, **f**, Kymographs of ablated junctions for the respective genotypes (each pixel = 2 s). **g** Average boundary recoil after ablation in *wt* control (green, n = 5), *nmo^mut^* (red, n = 7), and *nmo^GOF^* (blue, n = 8) tissues. Line shows mean, and shaded region indicates standard deviation. Exponential fitted curves for *wt* control and *nmo^mut^* are also shown. For *nmo^GOF^*, an initial expansion was immediately followed by reformation and shrinkage of the ablated boundary, and an exponential function does not correctly fit this data. **h** Table showing the initial recoil speed and recoil amplitude for *wt* control and *nmo^mut^* measured from the exponential fitted curves shown in (**g**). **i** Schematic summary of the effects of Nmo during rotation and the associated tissue fluidity requirements for rotation. Ommatidial cluster cells are colored in red and ICs in gray in all panels. The thickness of black lines between cells, representing AJs, correlates with levels of E-cad. Blue double arrows indicate tension levels, and red circular arrows the level of E-cad recycling. As E-cad recycling and tissue fluidity increase from *nmo^mut^*, through *wt* (control) to *nmo^GOF^*, tension decreases, facilitating cluster rotation in *wt* and ultimately allowing over-rotation in *nmo^GOF^*.

mesoderm invagination in *Drosophila*[55]. During anterior-posterior axis elongation in zebrafish, posterior tissues undergo a jamming transition: they are fluid-like at the extending end and become more solid in the presomitic mesoderm to support the extending tissue[12]. Actively-generated stress fluctuates, and this variability helps to fluidize the tissue at the posterior. Temporal and spatial regulation of jamming transitions are therefore critical to development and homeostasis; it has been proposed that organisms have evolved to occupy space around such transitions, which allow functional changes in tissue fluidity with relatively little effective energy expended[54]. Here we identify the developing

*Drosophila* eye tissue as occupying a similar space, with changes in Nmo levels regulating the balance around the transition point. At a phenomenological level, modeling based on shape index and cell packing is able to make accurate predictions about the level of tissue fluidity[55]. However, discovering the biological underpinning of this relationship, will require experimental manipulations of fluidity and thus Nmo provides a genetic tool to study the mechanics of tissues during morphogenesis in vivo.

Our data combined with previous observations suggest a multi-tiered requirement of Nmo in the rotation process[32,33]. The heterogeneous localization of core PCP factors within the cluster,

with Prickle and VanGogh localizing to the R4 side of the R3/R4 junction with their enrichment in R4 and their junctional recruitment of Nmo (ref. [33]), could imply that an initial anisotropic, polarized effect on junctions between the precluster cells and ICs defines the direction of ommatidial rotation, similar to neural plate morphogenesis[75]. Here the difference between junctional Nmo levels in the R3/R4 cell pair would provide a directional cue for rotation in the R4 direction. Subsequently and in parallel, the wider- and temporally-continuous requirement of Nmo in all R-cells and ICs[31,33,34] provides the tissue with the necessary fluidity, which in balance with the initial symmetry-breaking force generated by PCP signaling in the R3-R4 pair, regulates the extent of ommatidial rotation and allows its correct completion.

Our data further indicate that Nmo acts not only in the eye, but that its ability to regulate tissue fluidity and junctional dynamics is generally conserved in other tissues, such as the wing. The Nmo-like MAP kinase family (Nlk in vertebrates) is evolutionarily conserved, and its requirement has been linked to PCP-regulated morphogenetic processes in vertebrates as well, including cell movements during gastrulation[76]. Blastoderm spreading in the zebrafish relies on tight spatio-temporal control of tissue-fluidization[43]. This process of fluidization is spatially positioned by Wnt11/Fz7 PCP signaling and involves destabilization of E-cad contacts through a Rab5c/endocytosis-dependent mechanism[77]. The *nmo* homolog *nlk* is also required for convergent extension in zebrafish and strongly interacts with *wnt11/silberblick* during the process[76], and may operate in a similar manner as in *Drosophila* tissues, by regulating Wnt/PCP-induced E-cad turnover, which is planar polarized in both the germband[78] and the pupal wing[79].

Moreover, misregulation of Nmo-like kinases in humans, namely an increase in Nmo levels, has been correlated with various cancers[80,81], which together might suggest that Nmo generally regulates tissue fluidity and that dysregulation of tissue fluidity can have a negative impact in disease[54], including cancer[73]. The unjamming transition has been proposed to resemble invasion of interstitial spaces by tumor cells, particularly those that are not dependent upon epithelial-mesenchymal transition[56].

In summary, tissue fluidity has emerged as an important feature in several morphogenetic processes, ranging from body axis elongation to wound healing[12,82] and diseases, including asthma[54] and certain cancers[73], and thus understanding how fluidity is regulated at the molecular level is of great importance.

## Methods

**Fly strains, genetics, and histology.** *E-cad::GFP*[KI] is a knock in line generated by homologous recombination[83]. Both, the hypomorphic allele (*nmo*[P1]) and the null allele (*nmo*[DB]) were characterized in ref. [33]. All live imaging/movies representing the *nmo* mutant genotypes(s) were generated with a *nmo*[DB/P1] trans-allelic viable combination.

To generate the clonal tissue for *nmo*[DB] mutation the following strains were used:
*hs-FLP*[122]; *Ecad*::GFP[KI]; *His2Av mRFP,FRT79D/ TM6* and *Ecad::GFP*[KI]; *nmo*[DB],*FRT79D/ SM5a–TM6b*. F1 generation was heat shocked at 35 °C for 60 min on days 2 and 3 after egg laying.

*klarsicht* mutant line (*klar* or *klr*) *klr*[Δ118]/TM6B is a gift from Michael Welte. *zip::YFP* (myosin heavy chain fly TRAP line)[84], *mud*[RNAi] (BL38190 and BL28074), *mud*[1]/FM7,P[Act::GFP] (BL9562), *mud*[4]/FM4,P[ActGFP] (BL9563), UAS-Diap1(BL6657), *y,w,shi*[FL54] *P{ry[+t7.2] = neoFRT}19 A/FM7c* (BL51343), *w; P{w[+mC] = UASp-shg.R}5* (BL58494), and *w; P{w[+mC] = UASp-shg.ΔJM}3* (BL58444) were obtained from the BDSC (*shotgun/shg* is the *Drosophila* gene name for E-cad), and *Dp53*[RNAi] from the VDRC stock centers. Overexpression assays with *UAS-Nmo* (ref. [33]) used *sev-Gal4* (eye) and *ptc-Gal4* (wing) driver lines, as available from the BDSC.

**Live imaging.** Pupae were prepared as described in Supplementary Fig. 1 and imaged at room temperature (~22 °C) using a confocal scanning laser microscope, Zeiss LSM 780, 63X/1.4 oil-immersion objective with zoom ×3. Confocal Z series of 10−12 planes spanning 7−8 μm at the level of adherens junctions were acquired, tiles of 1×2 or 1×3 along the MF imaged at a rate of 1 frame every 10 min for 4–6 h were combined for analysis. In all movies, laser power and offset conditions were optimized to get the best signal while minimizing photobleaching.

**Image analysis and quantifications.** For the analysis of live microscopy movies shown in Figs. 1–4 and Supplementary Figs. 5, 6, images were first processed using a custom written macro in FIJI to correct nonlinear XYZ drift[85]. Crops of individual ommatidial pre-clusters, including their surrounding neighbor cells, were used to select the five best focus planes for a maximum-intensity projection and, if needed, further drift corrections were applied. These crops were chosen such that ommatidial clusters analyzed are far from each other and have no overlap. Therefore, measured events for each ommatidium are considered independent, both biologically and statistically. Drift-corrected movies were then segmented using the FIJI plugin, Tissue Analyzer[59]. The segmented images from each ommatidium were then imported to a custom-written MATLAB program to label and track cells and quantify various cellular properties per ommatidium. Temporal measurements from all ommatidia were aligned using the recruitment of R7 to the ommatidial cluster. For those ommatidia that R7 joined the cluster prior to the start of acquisition, the OM constriction curve was used to estimate R7 recruitment.

To quantify ommatidia rotation, the angle between the R2/R5 pairs relative to the morphogenetic furrow was measured as a function of time. To quantify ommatidial constriction, the combined apical surface area of R1−R8 cells were measured as a function of time. For each ommatidium, we tracked each of the first-level neighboring ICs (determined at the time of R7 recruitment and including cone cells) and scored whether there was an event (cell division, delamination, or neighbor exchange, respectively) between subsequent frames: $t$ to $t + 1$, $t + 1$ to $t + 2$ etc. For each frame, we calculated the number of ICs that had exhibited such an event. For cell division and delamination, we binned the measurements per ommatidium into bins of 90 min, and calculated the mean and standard error of the mean over different ommatidia. For neighbor exchange, in Fig. 2d, we calculated the mean and standard deviation of events per ommatidium, and in Fig. 3h, we calculated the mean and standard deviation of the fraction of cells that undergo neighbor exchange per ommatidium. To quantify cell displacement, cell centroids of interommatidial cells were tracked from the time of R7 recruitment up to 120 min after that as follows. For each IC, we measured the position of its centroid relative to the pre-cluster as a function of time (the pre-cluster is positioned in the origin). We then pulled the trajectories of all ICs from all OMs, divided the area around the origin into grids of $1.25 \times 1.25$ μm, and measured the displacement at each grid as the average displacement of ICs that pass through that grid. To quantify cellular speed in Figs. 2f and 3k, we used the displacement to calculate the speed on the grid points and then use interpolation to calculate speed on small grid size. The cell shape index in Fig. 4a, d, f, and i is calculated as $p_0 = L/\sqrt{A}$, where $L$ is the perimeter of the cell and $A$ is the cell area. To generate the graph in Fig. 4e, we calculate the mean and standard deviation of cell shape index of ICs per ommatidial field and plotted these for each ommatidium. The shaded region is the smallest ellipse that contains all the data points per genotype.

Adult eyes were embedded in resin[33] and rotation in the adult eye sections was quantified as the angle from the R1–R3 alignment, relative to the equator, using Photoshop.

**FRAP.** For the 10 μm Fluorescence recovery after photobleaching (FRAP) experiments were performed using Zeiss LSM 780 confocal microscope, objective 63X, zoom 6, 5 planes with 0.6 μm Z step were acquired at 20 s time intervals after photo-bleaching to avoid extensive bleaching during recovery time. An area (0.8 × 10 μm) was photobleached and laser power and iteration settings were adjusted to photo-bleach up to 80% of the initial intensity.

**FRAP quantifications.** To quantify the most meaningful data of the FRAP studies, we have focused on the rate of recovery and half recovery time. These are by far the most relevant information in the context of junctional stability and dynamics and it allows a direct comparison across different backgrounds. Due to the different steady-state junctional levels of E-cad (see Fig. 5a–c and Supplementary Fig. 7g, h) other measurements would not be very meaningful. The rate of recovery does not depend on absolute steady state protein levels.

To reduce the impact of intensity fluctuations due to changes of focal plane and natural acquisition bleaching, the intensity of the photobleached Region Of Interest (ROI) was corrected by measuring the fluctuations of intensity in unbleached/unaffected neighboring junctions and multiplying the intensity measured in the ROI by the coefficient of bleach. We performed quantifications on the sum of intensity projections of five focal planes spaced by 0.6 μm. The bleached region was tracked, and the mean intensity was measured using Fiji build in plug-ins[85]. The normalized fluorescence intensity, $I_N(t)$, was measured as

$$I_N(t) = (I(t) - I_{min})/(I_{max} - I_{min}) \quad (1)$$

where $I(t)$ is the intensity at time $t$, $I_{min}$ is the intensity immediately after bleaching and $I_{max}$ is the intensity before bleaching. The time $t = 0$ corresponds to the onset

of bleaching. The normalized intensity time traces were fitted, using MATLAB, to a single exponential as follows:

$$I_N(t) = A(1 - e^{-kt}) \qquad (2)$$

where $A$ is the mobile fraction, $t$ is the time after beaching and $k$ is a fit parameter. The recovery rate is calculated as $A \times k$, and the half recovery time, $T_{1/2}$, was calculated as follows:

$$T_{1/2} = -\ln(0.5)/k \qquad (3)$$

**Immunofluorescence microscopy on fixed tissues.** The primary antibodies used in this study from the Developmental Studies Hybridoma Bank: mouse anti-Elav antibody (Elav 9F8A9, 1:50), rat anti-Ecad (DCAD2, 1:20), mouse anti-Fmi antibody (Flamingo #74, 1:50), mouse anti-Dlg (4F3 Discs large, 1:200), and mouse anti-Ptc (APA1, 1:2); rabbit anti-Patj (1:1000) was a gift from Hugo Bellen. Secondary antibodies are from Jackson Immuno Research Laboratories and used at 1:300. Nuclear stain via Hoechst 33342 (10 mg/ml, 1:1000). Confocal images of fixed tissue were acquired on a Zeiss LSM 880 at the Microscopy CoRE of the Icahn School of Medicine at Mount Sinai.

**Laser ablation.** Pupae for laser ablation were prepared as for live imaging. For ablation of the adherens junctions, a custom-build system with a femtosecond near-infrared Ti:sapphire pulsed laser (Mai-Tai, Spectra-Physics, Mountain View, CA) combined with a pulse picker (Eclipse Pulse Picker, KMLabs) to generate 16 kHz femtosecond pulse train with ~6 nJ pulse energy was used. The imaging objective (CFI Plan Apo VC 60X WI, NA 1.2, Nikon) was used to focus the laser to a diffraction-limited spot, which then scanned over the sample in three dimensions at speed 40 μm/s to ablate adherens junctions by using a piezo-stage (P-545 PInano XYZ, Physik Instrumente) and home-developed LabVIEW software (LabVIEW, National Instruments). For imaging, an inverted spinning disk confocal microscope (Nikon Ti2000, Yokugawa CSU-X1) equipped with 488 nm laser and a Hamamatsu EMCCD camera was used.

To quantify junctional recoil after ablation, raw images were processed using a custom-made FIJI macro to first correct for the XY drift followed by generation of a kymograph along the ablated junction (between the two vertices). The distance between the two vertices, $l(t)$, was then measured from the kymographs[85]. The normalized recoil for each experiment was measured as $(l(t) - l_0)/l_0$, where $l_0$ is the length of the junction before the ablation, and the average recoil for each genotype was then quantified. For wt control and nmo LOF, an exponential curve, $L = l_f (1 - e^{-t/T})$, was fitted to the average recoil data. The initial recoil speed was measured as $l_f/T$, and the recoil amplitude was measured as $l_f$.

**Statistical methods.** Statistical analyses to determine p-values were the two-tail Student test for data in panels Figs. 3e, f, j, 4b–d, g–i, 5c, h, i and Supplementary Fig. 7f. To calculate the p-value for data displayed in panels Fig. 5k, and Supplementary Fig. 7b, c, we used the Two-sample Kolmogorov−Smirnov test.

Please note that the statistical analyses in most cases compare the distribution of values, not the averages, and as such the n values are either individual ommatidia, cells, or cellular junctions, consistent with many previous studies. For example, in the context of a distribution analysis, the n number is the number of individuals, either ommatidia or single cells in our case, depending on the measured parameter. Each ommatidium or cluster is independent from its neighbors in its phenotypic decision making in the PCP and ommatidial rotation context, as such the n value is the number of ommatidia analyzed in that context (and not the number of animals). It is not the average across animals that matters but the distribution of phenotypic values of individual ommatidia between genotypes. Similarly, when analyzing cellular behavior and cell shape, individual cells are the n value, as was described and used many times in the elegant and groundbreaking work of the Eaton and Julicher labs.

**Reporting summary.** Further information on research design is available in the Nature Research Reporting Summary linked to this article.

## Data availability
The data generated in this study have been deposited in the Figshare data repository under the following URL: https://figshare.com/articles/dataset/Data_Source/16864540. This URL is freely accessible with no restrictions. Source data are provided with this paper. The source data of experiments presented in this paper are organized in figure panel-specific Excel sheets and available at the above URL. Source data are provided with this paper.

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

## Acknowledgements

We are most grateful to Dan Needleman for his encouragement and help, and Hai-Yin Wu and Che-Hang Yu in the Needleman lab for allowing access to their laser ablation set up. We thank the Bloomington Stock Center, Michael Welte, and Esther Verheyen for fly strains, and the Developmental Studies Hybridoma Bank (DSHB, supported by the NICHD of the NIH and maintained at the University of Iowa), and Hugo Bellen for antibodies. We are grateful to Ivana Mirkovic for sharing unpublished results and all Mlodzik lab members for helpful input and many discussions, Jun Wu for technical advice, and C.P. Heisenberg and Benoit Dehapiot for helpful comments on the manuscript. This study was only possible thanks to the groundbreaking work of Suzanne Eaton, who has been inspirational to the field. This work was supported by NIH grants R21 HD095043 (technology development), R01 EY013256, and R35 GM127103 to MM; confocal microscopy was in part performed at the Microscopy CoRE, which is in part supported by the Tisch Cancer Institute P30 CA196521 grant from the NCI.

## Author contributions

N.F., R.F. and M.M. conceived and designed the study, also based on discussions with GMC. N.F., R.F., G.M.C. and U.W. performed experiments and/or analyzed the experimental data. N.F., R.F., G.M.C. and M.M. analyzed and interpreted the data and wrote the manuscript. M.J.S. and M.M. provided funding and oversight for the project.

## Competing interests

The authors declare no competing interests.
