## [Peer Review File · Nature Communications]

Tissue fluidity mediated by adherens junction dynamics promotes planar cell polarity-driven ommatidial rotationREVIEWER COMMENTS

Reviewer #1 (Remarks to the Author):

This study investigates the mechanism underpinning ommatidial rotation in the *Drosophila* developing eye disc epithelium. Here ommatidial clusters of initially 5 cells (later increasing to 8 cells) develop planar polarity and rotate by 90° within the epithelial layer. The authors develop a live imaging approach to observe the process and computational tools to evaluate movie sequences. The analysis shows that the rotation movement is pulsative and that periods of movement coincide with pulses of (presumably actomyosin driven) apical constrictions. The authors show that the rotation of the ommatidial clusters is not accompanied by a similar rotation of interommatidial cells suggesting that interommatidial cells act as a substrate for the (largely?) autonomous movement of the ommatidial cluster. They also show that cell divisions and cell delamination of interommatidial cells do not influence ommatidial rotation.

Previous work suggested that the MAP kinase Nemo controls ommatidial rotation. Confirming and expanding on these earlier findings, the author show through live analysis that rotation is slower and incomplete in *nemo* mutants and accelerated by *nemo* overexpression. It is also reported that neighbor exchange in the tissue in general is reduced in *nemo* mutants and increased by *nemo* overexpression. These findings suggest that Nemo modulates tissue fluidity, that is the degree, or ease of neighbor exchange, and therefore contributes to ommatidial rotation.

Much evidence from the literature suggest that tissue fluidity in epithelia is dependent on the dynamics of the cadherin-catenin complex (the more dynamic the complex is, the more fluid is the tissue). Moreover, previous work indicated that Nemo is recruited to adherens junctions and can phosphorylate E-cadherin and β -catenin. Supporting a downstream role of E-cadherin, the authors show that E-cadherin is more stable at the membrane in *nemo* mutant tissue. These enhanced E-cadherin levels correlate with enhanced junctional tension (and hence a stiffer tissue) as assessed via recoil analysis after laser cutting of junctions. In contrast, Nemo overexpression reduces tissue tension. Overall, the data suggest that Nemo regulates ommatidial rotation by regulating E-cadherin surface stability and the accompanying changes in tissue fluidity.

Overall, this is a well-executed study with a high level of technical competence. The concepts used (tissue fluidity and the dependency of it on E-cadherin mediated adhesion) are not novel. This is a nice example of how these ideas play out in a specific case, ommatidial rotation. As the fly eye is such an imported model tissue and the findings here will be embedded in a large amount of previous knowledge, I would think that the results are not only of interest to researcher of the fly eye, but to a broader audience.

I am concerned about the conclusion that the regulation of E-cadherin surface abundance through Nemo phosphorylation is the primary cause for tissue stiffening and regulation of fluidity which consequently regulates actomyosin (the force generating component in the system) is not well supported. While this model as presented in the discussion is consistent with the data, it is very much possible that Nemo regulates other components of the mechanical system that determines tissue stiffness such as the actin cytoskeleton. That modulation of E-cadherin levels at the membrane changes stiffness as predicted needs to be demonstrated through experimentation with E-cadherin to support the model and make the paper suitable for NatComm.

I noticed that some figure legends lack a complete set of statistical information including n value given for each experiment and tests employed. Statistical methods should also be described in the Method section.

Reviewer #2 (Remarks to the Author):

The manuscript presents new data on cell and tissue dynamics during ommatidial rotation (OR) in *Drosophila*, pointing toward a permissive role for the surrounding tissue in this rotation process. The authors present novel results on cell and tissue dynamics during OR, leveraging new insights from live imaging and quantitative analysis of this process. In particular, they present data supporting their model that the fluidity of the interommatidial cells (as measured by neighbor exchange and junctional dynamics) plays a permissive role in the rotation of clusters of ommatidial cells within the tissue. They show that the PCP effector protein Nemo modulates E-cadherin turnover and mechanical tensions at adherens junctions, which likely contribute to the observed changes in neighbor exchange and tissue fluidity in *nmo* mutants. These results add to a new and growing body of research revealing key roles for tissue fluidity in development and morphogenesis, in the interesting context of rotation of cell clusters relative to surrounding tissue. These findings are novel and timely and will be of interest to many readers of Nature Communication who study development and morphogenesis. The manuscript and results are generally clear and well-presented and most of the conclusions are supported by the presented data. However, there are two major concerns that somewhat dampen enthusiasm for the manuscript.

Major concerns:

- The authors argue that the behavior of tissues in wild-type, *nmo* loss-of-function, and gain-of-function pupae can be explained by regulated tissue fluidity, presumably by the surrounding interommatidial cells (ICs) providing the proper amount of mechanical resistance (friction) to rotation (with rotation driven autonomously by cells in the rotating cluster). Following from this, it seems that the boundary

between the rotating cells and the ICs would be critical in mediating this mechanical resistance from the ICs. However, the authors do not directly measure the junctional properties or dynamics at this boundary and instead focus on junctions between ICs (although this point is not entirely clear from the methods). Moreover, this picture seems difficult to square with the data presented in Fig 1i,j, which show very little correlation between the motion of the rotating cluster and the motions of the nearest neighbor ICs – what does this say about mechanical resistance from the surrounding ICs?

- In many figures, results are presented from a relatively small number of pupae and without much discussion of variability between pupae or between ommatidia. More detailed presentation and statistical analyses of these results, as well as additional examples from more pupae, would strengthen the manuscript.

Minor concerns and comments:

- The authors briefly touch on potential roles of basolateral integrin-based ECM attachment and cell migration in the discussion. An expanded discussion of this topic and a mention of this in the introduction would be helpful to the reader.

- The authors reference a small sampling of the growing body of literature on the roles of tissue fluidity, but seem to be missing some critical recent references. Referencing and discussing this literature in more detail would strengthen the manuscript and increase the impact of this work.

- The pulsing data in Figure 1c and 2h are interesting, but not addressed further in the discussion. Additional discussion of how the pulsing might be related to tissue fluidity would strengthen the manuscript.

- Figure 1f,h: Should there be error bars here? How many total cells were analyzed?

- Figure 1i,j: Is it correct that the data in j represent the average across many samples? If so, it would be helpful to also see examples from single ommatidia, e.g. in the extended data.

- Section on “ICs serve permissive role for OR progression”, final sentence. “Taken together our data suggest that the ICs serve a permissive role during the OR process”. The data in the section support the conclusion that the ICs do not play an instructive role. However, perhaps I missed something, but this section doesn’t directly show a role for ICs in OR. Perhaps it would be more accurate to say at this point in the manuscript that “... ICs serve at most a permissive role during the OR process”

- Figure 2a-f: corresponding images from controls would be helpful for direct comparison.

- Figure 2: Are the lines from segmentation overlaid on these images? If so, it would help for these lines to be in a different color.

- Figure 2g: Does this include only neighbor exchange between ICs, or does it also include exchanges between ICs and rotating cluster cells? How many total cells and junctions were analyzed? Should there be error bars here?

- Figure 2d,h: GOF data. Is the data shown for GOF in 2h consistent with the data in 2d showing that areas on average decrease similarly to in controls? Is there significant variability between ommatidial areas?
- Metrics for fluidity: In addition to using neighbor exchange as a metric for tissue fluidity, can cell shapes also be used to predict fluidity based on vertex or other physical models of epithelial tissues? For example, following work by Bi, Manning, Fredberg, and others?
- FRAP results: I'm not sure I follow the logic for explaining why in nmo mutants Ecad recovers to higher final levels than in controls. Shouldn't this result in higher steady state Ecad levels at junctions. This appears to be true in the pupal wing, but why not in the eye? Alternatively, could this be explained by a larger fraction of the total Ecad in cells being bleached in control compared to in mutant tissues? More explanation would be helpful.
- Figure 3a-c: Were FRAP measurements only done on junctions between two IC cells or also between an IC and a cluster cell? As mentioned above, it would be helpful to have information on junctional dynamics at the boundary between ICs and rotating clusters.
- Comment: Some of the roles of Nemo presented here are reminiscent of the roles of Abl kinase in regulating beta-catenin, junctional remodeling, and neighbor exchange during *Drosophila* germband extension (Tamada et al, Dev Cell 2012).
- Ablation experiments: Were measurements only done on junctions between IC cells or also between ICs and cluster cells? Also, did recoil behavior depend on position and orientation of the ablated junction relative to neighboring ommatidia?

Reviewer #3 (Remarks to the Author):

This manuscript is the first (to my knowledge) to use a live imaging approach to study ommatidial rotation in the *Drosophila* eye. Much is known about e.g. convergent extension in epithelia, but study of more complex processes such as ommatidial rotation gives the possibility of new insights.

The basic story is simple and interesting. The data support a model in which ommatidia undergo continuous rotation for 90° (rather than in two 45° steps as suggested by work in fixed tissue) apparently due to forces generated in the ommatidial cells themselves (the data hint at actomyosin pulses as seen in T1 transitions, but this isn't followed up). Meanwhile, the ability to rotate is modulated by the surrounding interommatidial cells: if these have high cortical tension and low neighbor exchange then rotation is inhibited, whereas if they have low cortical tension and high neighbor exchange then rotation is enhanced (and goes too far). Furthermore, the kinase Nmo controls cortical tension and hence presumably neighbor exchange in the interommatidial cells.

However, regarding the mechanism of action of Nmo, I found the data puzzling. It appears that total E-cad levels don't change in Nmo loss-of-function and gain-of-function conditions, and FRAP shows that both conditions reduce the stable fraction of E-cad at junctions, so both conditions are expected to reduce cell adhesion. Yet, Nmo loss-of-function and gain-of-function conditions have opposite effects on ommatidial rotation and on junctional tension. I'm not sure if I missed something, or the authors missed something. However, regardless of the interpretation of the FRAP, there doesn't seem to be any direct evidence suggesting the mechanism is partly or entirely dependent on E-cad mediated cell adhesion and not on effects on actomyosin and the cortical cytoskeleton.

I've made suggestions for how the authors could address some of these issues, however the fundamental question of roles of adhesion vs tension probably requires some investigation of effects of Nmo on the actomyosin cytoskeleton.

Minor points:

First sentence of abstract - "the molecular basis of tissue fluidity remains unknown" seems a bit of a sweeping statement. In some cases a good case can be made for processes such as cell intercalation (as in ref. 1) underlying fluidity. So maybe this statement should be qualified in some way? (Or removed?)

Second sentence of abstract - "OR": I tripped over this (new?) abbreviation every time I read it. Would it increase readability to simply say "ommatidial rotation" instead? Hopefully space is not so limited that this would be an issue. (The same might be said of "MF" and "IC", at least for readers not familiar with eye development.)

Second sentence of abstract - how similar is ommatidial rotation to cell migration? I think ommatidial rotation occurs in an epithelium where cells retain adhesive junctions, while migrating cells undergo EMT thus significantly altering their properties. I suspect the authors may be trying to make their favorite model sound more broadly relevant, but I don't think this is necessary (and if it is, what about an example of cell intercalation in a vertebrate epithelium?).

Final sentence of abstract - "Our study defines Nemo as a molecular tool" is quite ambiguous. Do the authors mean a tool for investigators to use (i.e. the conventional sense) or do they mean "the tissue" uses Nemo as a "tool"? Is it easier to say "modulation of Nemo activity is a molecular mechanism"?

p.3 - "it remains largely unknown how fluidity of a tissue is regulated at the molecular level": this statement might come as a shock to the many other labs studying cell intercalation in Drosophila (Zallen, Lecuit, Bellaiche, Eaton and many others). Regulation of E-cad and actomyosin come to mind as the major mechanisms (with I think a number of studies at stages where proliferation is absent or sufficiently low to not play a major role?).

p.5 - "yet cluster composition and rotation direction remain unaffected": can we be sure of this by just looking at photoreceptor markers in eye discs or in adult eye sections? I agree the clusters look "normal" but has anyone ever looked at e.g. cone cells or other cell types recruited subsequently?

p.6 - "Consistently with this data, inhibition of cell delamination did not affect OR": I agree the clusters achieve the correct degree of rotation based on the adult eye sections, but it's not obvious this demonstrates that the process of ommatidial rotation is completely normal (it could for instance be

delayed and then catch up). The same applies to oriented cell divisions. So maybe these statements could be toned down?

Is SV2 ever cited in the text?

p.7 - "Taken together our data suggest that the ICs serve a permissive role during the OR process." I agree with the general conclusion based on the findings thus far. However, the rest of the manuscript goes on to show that different interommatidial cell behaviors result in different degrees of rotation, which seems like an instructive role?

p.8/Fig.2D and H - in Fig.2D the averaged values show Nmo GOF showing pretty similar average ommatidial area to wt, yet the example of an individual ommatidium in Fig.2H gives quite a different profile. Presumably to get the average profile seen in Fig.2D, some individual ommatidia would have to contract faster than wt, if others are contracting more slowly?

p.9 - "whereas nmo GOF revealed enhanced cell displacement": doesn't Fig.2L show no difference in cell displacement for Nmo GOF? I'm also not sure "velocity" is increased relative to wt as in Fig.S4B it seems to start rather lower than wt and then converge to a wt level. The only evidence for Nmo GOF boosting neighbor exchange seems to be Fig.2G (plus possibly the delamination data)?

p.10 - "E-cad fluorescence recovery can be achieved either through lateral movement within the plasma membrane or by recycling from the vesicular/endosomal compartment" or (for completeness) by delivery of newly synthesized protein?

p.10 - "While no difference was observed in the first experimental set up between the three genotypes (Extended Data Fig. 6e-h)": actually only show wt and nmo LOF? Is nmo GOF the same? I'm also surprised the authors conclude wt and nmo LOF are the same. My understanding is that without an a priori power calculation, just looking at the p-value doesn't tell you that the populations are not different.

Fig.3 and Fig.S6 - would it be better to put both nmo LOF and GOF FRAP in the main figure (on the same graphs)? This would make comparison easier for the reader.

p.10 - the authors' interpretation of the FRAP data seems to depend on the assumption that all recovery is via recycling. Not only does this ignore the possibility of diffusion (which doesn't seem to be ruled out as a mechanism) but also of new synthesis. It would seem simpler at this stage in the story to stick to the basic message that nmo LOF shows lower flux of mobile E-cad and nmo GOF shows higher flux of mobile E-cad without presupposing the exact mechanism? (Although I'm unsure this measure is actually a useful one, see next comment.)

p.10 (last words) - "The pupal wing..." actually should say "prepupal" as indicated in Methods? Also p.11 3rd line from bottom "The Drosophila pupal wing", should this also be prepupal wing? The text implies measurements of apical area and hexagonal packing (Fig.3G-H) are all in pupal wings but without indicating the age, however earlier it is suggested that Fig.3F,G are pulse-chase data in prepupal wings? (Actually in the preceding section it is often hard to know if the data being described is pulse-chase as it says things like "E-cad antibody at AJs exhibit a higher junctional intensity in nmo mutant cells, as compared to controls" which could be taken to mean steady state not pulse-chase measurements.) Reworking this section of the Results to be more explicit would be very helpful.

p.10-11 - I agree that the E-cad antibody uptake experiment is the right one to measure rates of E-cad internalization from junctions (albeit I'm not sure how much this tells us about strength of adhesion), but if this is relevant shouldn't this be done in eye discs? It is not obvious that Nmo function will be the same in wings. Such assays have been published in wing discs where apparently the peripodial membrane is sufficiently ruptured during dissection and I would think the same was certainly true at the back of the eye disc.

- Also, as stated in the text, to show the change is due to internalization you need to quantify levels that appear in cytoplasmic vesicles (preferably showing they are Rab5 positive) but the authors say that strong junctional labeling prevents this, but such vesicles would not be expected to overlap with the junctions in xy or z, which maybe indicates that they are actually not seeing significant internalization over the course of the experiment? Furthermore, to be sure there is actually any change in junctional levels, we need to see a timecourse (i.e. no internalization, 5 min, 10 min etc)

- And what happens in nmo GOF, do you see the expected increase in flux of the mobile E-cad fraction?

- Perhaps the simplest thing for the authors would be to remove the current data from the manuscript? A better use of the space might be to show steady state E-cad levels (possibly in permeabilised vs unpermeabilised, but this is less important) in wt, LOF and GOF eye discs? (See major point below.)

Although the authors mention it a few times (e.g. in the title or statements such as "Nmo levels directly affect AJs downstream of PCP") I found it hard to see the relationship to PCP, which seems to have a separate role in cell fate specification and determining direction of rotation, rather than directly regulating Nmo (e.g. spatial patterns of Nmo activity). Mirkovic et al 2011 seems to imply Vang/Pk localize Nmo activity (at the R3/R4 boundary?) but I don't see how this fits the authors' current model for the role Nmo in ommatidial rotation. Isn't this all about Nmo function outside the photoreceptor cluster, with no obvious role for planar cell polarized Nmo activity? And isn't the same true in the wing? Otherwise you would expect planar polarized differences in junctional tension (which would be easy to measure). Or are Vang/Pk playing a non-PCP role in generally recruiting Nmo to membranes (in which case it isn't strictly speaking "downstream of PCP" but "downstream of PCP factors playing a non-PCP role"?).

Major points:

p.10 - There is an anomaly in the FRAP data which seems hard to explain, which is that both nmo LOF and GOF result in higher mobile fractions of E-cad at junctions. If total E-cad levels remain constant at junctions in nmo LOF and GOF (which I think these authors published previously?) then both nmo LOF and GOF will have less immobile (i.e. bound) E-cad at junctions and thus will be expected to have weaker adhesion. This seems to be at odds with the model of the authors that nmo LOF has more stable junctions with fewer neighbor exchanges while nmo GOF has less stable junctions with more neighbor exchanges. I don't think the rate of turnover of the mobile fraction (which the authors seem to focus on) is the relevant measure here, except in as much as it might contribute to the size of the immobile fraction (a relationship that seems murky) and thus inferred strength of adhesion.

p.11 - "This confirms our interpretation of the FRAP experiments that the higher recovery level post-bleach would result in increased E-cad steady state levels at AJs." (See also section heading on p.9 "Nmo regulates E-cad levels at AJs".) I'm not sure if I missed something, but aren't steady state levels the same

in wt, nmo LOF and nmo GOF? On p.5 it is stated “nmo does not affect the overall E-cad levels” and I don’t recall seeing new data in the manuscript to contradict this. Changes in steady-state levels would help to explain some of the puzzling aspects of the data, so maybe this should be re-investigated?

p.12 - the laser ablation experiments give a pleasingly clear result, showing increased junctional tension in nmo LOF and decreased in nmo GOF. However, I don’t see why the authors think this shows this is (entirely) due to changes in E-cad mediated adhesion. On p.13 the authors say “our data indicate that the control of junctional E-cad levels and dynamics can directly modulate tissue tension, viscoelasticity, and stress, thus affecting tissue fluidity and cell motility” but this seems at best speculation. As noted above, nmo LOF and GOF seem to both reduced levels of immobile E-cad at junctions, so both would be expected to have decreased adhesion. If this is true, it seems rather more likely that Nmo is acting by modulating actomyosin and thus cortical tension directly, with LOF and GOF having opposite effects.

REVIEWER COMMENTS

Point-by-point response outline to each comment

Our responses are marked in red

Reviewer #1 (Remarks to the Author):

This study investigates the mechanism underpinning ommatidial rotation in the *Drosophila* developing eye disc epithelium. Here ommatidial clusters of initially 5 cells (later increasing to 8 cells) develop planar polarity and rotate by 90° within the epithelial layer. The authors develop a live imaging approach to observe the process and computational tools to evaluate movie sequences. The analysis shows that the rotation movement is pulsative and that periods of movement coincide with pulses of (presumably actomyosin driven) apical constrictions. The authors show that the rotation of the ommatidial clusters is not accompanied by a similar rotation of interommatidial cells suggesting that interommatidial cells act as a substrate for the (largely?) autonomous movement of the ommatidial cluster. They also show that cell divisions and cell delamination of interommatidial cells do not influence ommatidial rotation.

Previous work suggested that the MAP kinase Nemo controls ommatidial rotation. Confirming and expanding on these earlier findings, the author show through live analysis that rotation is slower and incomplete in *nemo* mutants and accelerated by *nemo* overexpression. It is also reported that neighbor exchange in the tissue in general is reduced in *nemo* mutants and increased by *nemo* overexpression. These findings suggest that Nemo modulates tissue fluidity, that is the degree, or ease of neighbor exchange, and therefore contributes to ommatidial rotation.

Much evidence from the literature suggest that tissue fluidity in epithelia is dependent on the dynamics of the cadherin-catenin complex (the more dynamic the complex is, the more fluid is the tissue). Moreover, previous work indicated that Nemo is recruited to adherens junctions and can phosphorylate E-cadherin and β -catenin. Supporting a downstream role of E-cadherin, the authors show that E-cadherin is more stable at the membrane in *nemo* mutant tissue. These enhanced E-cadherin levels correlate with enhanced junctional tension (and hence a stiffer tissue) as assessed via recoil analysis after laser cutting of junctions. In contrast, Nemo overexpression reduces tissue tension. Overall, the data suggest that Nemo regulates ommatidial rotation by regulating E-cadherin surface stability and the accompanying changes in tissue fluidity.

Overall, this is a well-executed study with a high level of technical competence. The concepts used (tissue fluidity and the dependency of it on E-cadherin mediated adhesion) are not novel. This is a nice example of how these ideas play out in a specific case, ommatidial rotation. As the fly eye is such an imported model tissue and the findings here will be embedded in a large amount of previous knowledge, I would think that the results are not only of interest to researcher of the fly eye, but to a broader audience.

We thank the reviewer for their nice summary of our manuscript and for their positive assessment of our work. It is gratifying to read comments like “well-executed study”, and a statement that our study is of interest to a broad audience.

I am concerned about the conclusion that the regulation of E-cadherin surface abundance through Nemo phosphorylation is the primary cause for tissue stiffening and regulation of fluidity which consequently regulates actomyosin (the force generating component in the system) is not well supported. While this model as presented in the discussion is consistent with the data, it is very much possible that Nemo regulates other components of the mechanical system that determines tissue stiffness such as the actin cytoskeleton. That modulation of E-cadherin levels at the membrane changes stiffness as predicted needs to be demonstrated through experimentation with E-cadherin to support the model and make the paper suitable for NatComm.

We thank the reviewer for highlighting their concern and making constructive suggestions to help further improve the paper. We have addressed the point raised in multiple ways:

- We have included a greater discussion of the relevant literature to show that, although actomyosin activity is required for the rotation process as would be expected for cell movements, the myosin LOF phenotype is significantly different to *nmo* LOF defects. In LOF mutants of ROK/myosin activity, the apical surface fails to contract in addition to the rotation defects (as was previously shown: Escudero, Bischoff and Freeman, Dev Cell 2007). Moreover, work from the lab of Tanya Wolff has shown that although *Nmo* and *Zipper* both affect rotation, they act in different genetic pathways (Fiehler and Wolff, Dev Biol. 2007). We also show that *Zipper* localization is not affected in the *nmo* null clone cells (Ext Data Fig 4f).
The phenotype of *NmoGOF*, however, showing extended periods of expansion, is similar to the published phenotype of *Rab35*, an endocytosis/trafficking factor, which affects apical membrane constriction by failing to stabilize pulses (Jewett et al, Nature Communications 2017), as does *NmoGOF* in the eye. We have added discussion of these comparisons throughout the text where appropriate both in the results and discussion sections.
- We have documented the change in steady state E-cad levels (new Figure 5) both in *nmo* mutant tissue as well as in *NmoGOF*, showing that less stiff tissue has less steady-state E-Cad at the junctions. This is now analyzed in detail in both *nmo* LOF and GOF tissue, showing that *Nmo* protein/activity levels inversely correlate with E-cad levels at the junctions.
- We have investigated the phenotypic effect of endocytic recycling on the *sev>Nmo* genotype (*NmoGOF*), and show that loss of function of dynamin (*shibire* in *Drosophila*), which should reduce E-cad recycling from membrane and thus increase steady state levels, indeed suppresses the *Nmo* GOF phenotype of rotation defects. Similarly, overexpressing an E-cad molecule that cannot bind to p120-catenin, and thus should be trafficked more easily, has a stronger enhancement of *Nmo* GOF rotation phenotype than does wt E-cad (labelled *shotgun* [the gene name in *Drosophila*] in the legend). These new added data are shown in the new Figure 5 and Extended Data Fig. 7. Together these data strongly argue for a role of *Nmo* in regulating the dynamics of junctional E-cad.
- In further support of our conclusion, we have added a whole new figure (new Figure 4) on tissue fluidity and how it is linked to the “cell shape factor” (as defined and used by the Bi, Fredberg and Manning labs). These new data demonstrate the effect of *Nmo* on several cell size/shape parameters and connect it well to existing assays and literature on tissue fluidity and tissue jamming/unjamming transitions. Consistent with these new data of increased fluidity, existing literature shows that E-cad trafficking - but not actomyosin contractility - affects tissue fluidity and a tissue jamming/unjamming transition. We have also added description and discussion of these features to the new section of the Discussion (please see references there).
- In this context it is also worth noting that the phenotype of *nmo* is very similar to effects of p120-catenin and other junctional dynamics regulators in wing tissue, as was shown in work from the Eaton lab (for example Classen et al 2005, Iyer et al 2019). Our new data is presented in the new Figure 4, showing both data in wing tissue and the eye.

We have expanded the discussion of each of these points to make it clearer and easier to follow for the reader. This is in addition to also expanding discussion of the literature in the Wnt/PCP signaling-E-cad trafficking context and its suggested links to tissue fluidity. We trust that the addition of the new data and further analyses and the clarification of the above-mentioned issues improve the paper further and make it easily accessible to a general readership.

I noticed that some figure legends lack a complete set of statistical information including n value given for each experiment and tests employed. Statistical methods should also be described in the Method section. We thank the reviewer for pointing this out and apologize an omission of some of the statistical details. This has been now addressed, and the statistical methods are described in the revised methods section. Also, the figure legends include n and p values as appropriate.

Reviewer #2 (Remarks to the Author):

The manuscript presents new data on cell and tissue dynamics during ommatidial rotation (OR) in *Drosophila*, pointing toward a permissive role for the surrounding tissue in this rotation process. The authors present novel results on cell and tissue dynamics during OR, leveraging new insights from live imaging and quantitative analysis of this process. In particular, they present data supporting their model that the fluidity of the interommatidial cells (as measured by neighbor exchange and junctional dynamics) plays a permissive role in the rotation of clusters of ommatidial cells within the tissue. They show that the PCP effector protein Nemo modulates E-cadherin turnover and mechanical tensions at adherens junctions, which likely contribute to the observed changes in neighbor exchange and tissue fluidity in *nmo* mutants. These results add to a new and growing body of research revealing key roles for tissue fluidity in development and morphogenesis, in the interesting context of rotation of cell clusters relative to surrounding tissue. These findings are novel and timely and will be of interest to many readers of *Nature Communications* who study development and morphogenesis. The manuscript and results are generally clear and well-presented and most of the conclusions are supported by the presented data.

We thank the reviewer for their positive assessment of our study and data, referring to our work as “novel and timely” with “clear and well presented” results.

However, there are two major concerns that somewhat dampen enthusiasm for the manuscript.

Major concerns:

- The authors argue that the behavior of tissues in wild-type, *nmo* loss-of-function, and gain-of-function pupae can be explained by regulated tissue fluidity, presumably by the surrounding interommatidial cells (ICs) providing the proper amount of mechanical resistance (friction) to rotation (with rotation driven autonomously by cells in the rotating cluster). Following from this, it seems that the boundary between the rotating cells and the ICs would be critical in mediating this mechanical resistance from the ICs. However, the authors do not directly measure the junctional properties or dynamics at this boundary and instead focus on junctions between ICs (although this point is not entirely clear from the methods).

We appreciate this comment and associated suggestions. We have addressed this throughout the paper as needed.

The reviewer is correct that the boundary between rotating cells and ICs is a site of mechanical resistance, as has been shown by the progressive increase in Zipper localization at the boundary, when each new cell is added to the cluster (Fiehler and Wolff *Dev Biol* 2007). While the boundary between photoreceptor cells (PRs) and ICs is important, technical reasons mean these specific junctions are very difficult to address experimentally and the interpretation of such experiments might be also almost inconclusive (see below). In addition, patches of mutant ICs lacking Ecad can affect the rotation of wt neighboring clusters, which suggests that other IC behavior besides adhesion complexing affect cluster rotation (Mirkovic & Mlodzik, *Development*, 2006). We have included discussion of both these points in the updated text.

The junctions between the 1st neighbor ICs and their own neighbors represent the general fluidity of the tissue and also influence the ability of the cluster to rotate. We have expanded our discussion of tissue fluidity to emphasize that it is the surrounding environment as a whole that influences how a cluster rotates, and not just the specific adhesive contacts between the PR cluster and the static 1st neighbor ICs. We show that the IC speed correlates with degree of rotation in new data added as a heat map to the revised Figure 3 (formerly Fig. 2), shown in panels in Fig 3k. We also show in the revised paper that Nmo affects generally tissue fluidity (as determined via cell shape index measures) in wing tissue as well as the eye, and thus conclude that Nmo generally affects tissue fluidity (new Figure 4).

The cluster itself is not spherical and so as it rotates, there must be (general) movement of ICs as they are displaced – consider a square rotating vs a circle; even if the perimeter is the same length with the same amount of adhesion complexes, for the square or polygon to rotate it has to displace surrounding tissue as the corners move past it (also comparable to a paddle wheel being the cluster and the surrounding water mass being the ICs, as such surrounding general fluidity is important – if the water were changed to honey, the paddle wheel would rotate less using a given force).

We have included discussion of the importance of the boundary, and have expanded upon our discussion of fluidity as generally controlled by *Nmo*. In this context, we have not only added a whole new figure but also revised the new Fig. 5 with additional data [formerly Fig. 3]).

We trust that the reviewer will agree with us that the new data added and the expanded explanations make this important point and conclusion clearer.

[Technical difficulties and concerns of addressing R-cell/IC junctions:
The respective junction length is small, around 1-2um only (due to PR cluster contraction) and the laser ablation line of 0.5um is just too close to that. Also, the behavior/property of the ommatidial PR cluster is likely to affect the outcome, with ongoing constriction and very tight adhesion between the PRs themselves (as deduced from the highly upregulated E-cad levels in PRs). These features do interfere with a clean interpretation of junctional expansion,]

Moreover, this picture seems difficult to square with the data presented in Fig 1i,j, which show very little correlation between the motion of the rotating cluster and the motions of the nearest neighbor ICs – what does this say about mechanical resistance from the surrounding ICs?

Again, a very important point, and we apologize, if we failed to explain this adequately.

In the revised Figure 2 (panel 2f) we document that the direction of rotation does not correlate with the displacement vector of rotation, i.e. that the surrounding ICs are not pulling along the PR cluster with them. What we demonstrate (new panel in the new/revised Figure 3) is that the displacement and speed of ICs fully correlate with the level of rotation. Panels in the new Figure 3k, displaying this as heat maps, demonstrate the speed of movement of surrounding ICs to make this clearer. Also, the cell centroid displacement is very much reduced in the *nmo* mutant background (shown in panel in new Fig. 3j), consistent with the ICs in *nmo* LOF being caged (see also individual tracking of IC centroids in Fig. 3i and Extended Data Fig. 5h), and thus unable to accommodate rotation of the PR cluster.

Further, as we show in Fig. 1g (and also Extended Data Fig. 6d-f), ommatidial rotation happens in pulses. These pulses of rotations generate local stresses in the surrounding tissue and among the ICs. If the ICs were completely immobile, similar to the LOF condition, these stresses will accumulate and cause increasing resistance to the ommatidia rotation. However, mobility of the ICs, resembling in net cell movement and cell neighbor exchange, can reduce these stresses caused by ommatidial rotation, and thus facilitate the rotation. The movement of the ICs might not be directly caused by the rotation of ommatidia, and thus not correlate with rotation, but the mobility of the ICs is important for stress relaxation caused by ommatidial rotation and thus the progression of the rotation process.

Along these lines, the IC motion correlates with tissue fluidity values, as shown in the all new Fig 4, which demonstrates this in both wings and eyes via cell shape factor calculations

In addition to the new data and figures, we have rewritten the discussion and included a section discussing tissue fluidity in general to explain this further.

- In many figures, results are presented from a relatively small number of pupae and without much discussion of variability between pupae or between ommatidia. More detailed presentation and statistical analyses of these results, as well as additional examples from more pupae, would strengthen the manuscript.

We apologize for not making this clearer. We have included a whole new figure (Extended Data Fig 6) showing the similarity in phenotype of different pupae of the same genotype (Ext Data Fig 6g-l), and also between ommatidia within same pupa.

We have included additional examples for cell rotation and constriction pulses, and centroid trajectories for all genotypes in the Extended Data figure set to also show multiple individual examples (i.e. Ext Data Fig 5h for centroid trajectories, and Ext Data fig 6a-f, compare to 1g and 3d in main text).

Minor concerns and comments:

- The authors briefly touch on potential roles of basolateral integrin-based ECM attachment and cell migration in the discussion. An expanded discussion of this topic and a mention of this in the introduction would be helpful to the reader.

We thank the reviewer for this suggestion. We had minimized this due to length restrictions of the original submission (letter format at Nature). We have now expanded upon this, both in the introduction and the discussion. We have also included new data in Ext Data fig 7b, showing the genetic interaction between *nmoGOF* and *myospheroid/integrin*.

- The authors reference a small sampling of the growing body of literature on the roles of tissue fluidity, but seem to be missing some critical recent references. Referencing and discussing this literature in more detail would strengthen the manuscript and increase the impact of this work.

We apologize for the limited set of references, again caused by the limits of the original submission format. Accordingly, we have now greatly expanded our coverage of the literature and importantly also added a whole new data set (new Figure 4) using methods described within the newly added literature to demonstrate tissue fluidity differences between the respective genotypes using multiple measures, firmly indicating that an increase in Nmo levels correlates with an increase in fluidity in both the eye and wing tissue.

- The pulsing data in Figure 1c and 2h are interesting, but not addressed further in the discussion.

Additional discussion of how the pulsing might be related to tissue fluidity would strengthen the manuscript. Thank you for this supportive comment. We have added discussion of the pulsing and how it compares to actomyosin pulsing in embryonic convergence extension and gastrulation processes. In addition, we have added more examples of the rotation pulsing to the new Extended Data fig 6 (panels 6d-f).

- Figure 1f,h: Should there be error bars here? How many total cells were analyzed?

We thank the reviewer for pointing this out – we have added error bars to both figures. The measurements were averaged over 11 independent clusters from 2 pupae in movies of >300mins. This information is now added also in the figure legend.

- Figure 1i,j: Is it correct that the data in j represent the average across many samples? If so, it would be helpful to also see examples from single ommatidia, e.g. in the extended data.

The data in the figure (now panel 2f in the new Figure 2) is an average of all the ommatidial clusters examined. We have changed the presentation of this data (as a heat map) to make it clearer. The details are explained in more detail in the methods. Also, as requested individual examples are now shown in Fig 2e and 3i, and also in the Extended Data Fig. 5h.

- Section on “ICs serve permissive role for OR progression”, final sentence. “Taken together our data suggest that the ICs serve a permissive role during the OR process”. The data in the section support the conclusion that the ICs do not play an instructive role. However, perhaps I missed something, but this section doesn’t directly show a role for ICs in OR. Perhaps it would be more accurate to say at this point in the manuscript that “... ICs serve at most a permissive role during the OR process”

We have rewritten this section and it now concludes that “ICs may serve a permissive role”

- Figure 2a-f: corresponding images from controls would be helpful for direct comparison.

We have rearranged the figures and included an extra example of a control ommatidium for each case (new Figure 3a,g and i).

- Figure 2: Are the lines from segmentation overlaid on these images? If so, it would help for these lines to be in a different color.

Thank you for this suggestion. All panels showing cells have now the same color for the overlaid

segmentation. We have chosen white as the most neutral outline as it least interferes with the color coding of the individual cells highlighting the respective features being analyzed (we tried other colors but these made the figures either too noisy or interfered with the interpretation/presentation of colors representing the specific features analyzed). This applies to all figures that display segmented images across the paper.

- Figure 2g: Does this include only neighbor exchange between ICs, or does it also include exchanges between ICs and rotating cluster cells? How many total cells and junctions were analyzed? Should there be error bars here?

The data include any neighbor exchange for the first row of IC neighbors: exchanges between ICs and rotating cluster cells are included. We have replotted the data and include error bars. The data are averaged from 17 independent mutant and 18 GOF ommatidia, respectively, from 2 pupae. We have added this detail to the figure legend.

- Figure 2d,h: GOF data. Is the data shown for GOF in 2h consistent with the data in 2d showing that areas on average decrease similarly to in controls? Is there significant variability between ommatidial areas?

We have added an extra set of examples in Ext data fig 6 – d-f. The variability is shown by the cloud of SD (now in figure 3c), and, although there is variation, the overall trend is the same. See also quantification of expansion and contraction in Fig 3e-f.

- Metrics for fluidity: In addition to using neighbor exchange as a metric for tissue fluidity, can cell shapes also be used to predict fluidity based on vertex or other physical models of epithelial tissues? For example, following work by Bi, Manning, Fredberg, and others?

We thank the reviewer for this excellent suggestion. We have now applied the equivalent measurements as outlined in work from Bi, Fredberg, and Manning among others, and used these to quantify both the average shape index, and the standard deviation of the shape index, both of which demonstrate increased fluidity in the *Nmo* GOF and decreased fluidity in the *nmo* mutant. Accordingly, this suggests that *Nmo* regulates a jamming/unjamming transition. This additional data has been added in the all new Figure 4, and is discussed in the results and discussion sections accordingly.

In addition, we have discussed the polygon classes, and compared these features to published literature related to tissue fluidity and jamming (i.e. as also outlined in the wing tissue by Classen et al 2005, and Iyer et al 2019).

- FRAP results: I'm not sure I follow the logic for explaining why in *nmo* mutants *Ecad* recovers to higher final levels than in controls. Shouldn't this result in higher steady state *Ecad* levels at junctions. This appears to be true in the pupal wing, but why not in the eye? Alternatively, could this be explained by a larger fraction of the total *Ecad* in cells being bleached in control compared to in mutant tissues? More explanation would be helpful.

Thank you for these suggestions on how to clarify this point.

We have added new data on steady state levels of E-cad in both *nmo* mutant and GOF backgrounds at a developmental stage of a homogenous cell population that is commonly used for such comparisons, pupal wings at 22h APF (see also Iyer et al 2019 from the Eaton lab for example). This allows not only the comparison of junctional E-cad levels in the different *nmo* backgrounds (New Figure 5a-c and Ext Data figure 7e-h) - and yes, the junctional steady state levels of E-cad do differ between the three genotypes - but it also lends itself for a direct comparison to other related work by the Eaton lab (e.g. Classen et al 2005 and Iyer et al 2019). Along these lines, the wing tissue is more suited for such analyses as eye tissue behind the furrow is too dynamic, as it is a temporal progression of development with cells adopting different fates and thus the steady state of E-cad there is difficult to compare.

Importantly, we have also redone the FRAP experiment analyses and now compare all three genotypes accordingly. This is now shown in the revised/new Figure 5 (panels 5g-l; and also Ext. Data Fig. 7a-d). We also explain this in detail in the text (see also response to the critique from reviewer 3).

- Figure 3a-c: Were FRAP measurements only done on junctions between two IC cells or also between an

IC and a cluster cell? As mentioned above, it would be helpful to have information on junctional dynamics at the boundary between ICs and rotating clusters.

The junctions between PRs and ICs are too small to do a reproducible FRAP analysis. We have hence applied this to the next neighbor junctions of the ICs. The new data analyses – for all three genotypes – are presented in the new panels Figure 5g-l, and this is also aligned with a genetic experiment using dynamin (*shibire* in *Drosophila*) and p120-catenin to document a requirement of dynamin/endocytosis in the process (new Fig. 5j-k; and Extended Data Fig 7b-c).

- Comment: Some of the roles of Nemo presented here are reminiscent of the roles of Abl kinase in regulating beta-catenin, junctional remodeling, and neighbor exchange during *Drosophila* germband extension (Tamada et al, Dev Cell 2012).

Thank you for this observation and suggestion. Yes, the roles of the two kinases are similar, and we have included the reference to the Tamada et al paper accordingly.

- Ablation experiments: Were measurements only done on junctions between IC cells or also between ICs and cluster cells? Also, did recoil behavior depend on position and orientation of the ablated junction relative to neighboring ommatidia?

This is correct and related to the major concern #1 above. Please see our detailed comments and adjustments to this, as discussed above. Our data do not suggest that the orientation of an ablated junction matters in the eye tissue. However, as the junctional orientation in the eye is much more random than for example in the pupal wing (where differences have been seen), we do not feel that we have enough data to make a firm conclusion about the effect of position and orientation of ablation on recoil behavior in the eye, and hence we do not wish to make a strong point about it.

Reviewer #3 (Remarks to the Author):

This manuscript is the first (to my knowledge) to use a live imaging approach to study ommatidial rotation in the *Drosophila* eye. Much is known about e.g. convergent extension in epithelia, but study of more complex processes such as ommatidial rotation gives the possibility of new insights.

The basic story is simple and interesting. The data support a model in which ommatidia undergo continuous rotation for 90° (rather than in two 45° steps as suggested by work in fixed tissue) apparently due to forces generated in the ommatidial cells themselves (the data hint at actomyosin pulses as seen in T1 transitions, but this isn't followed up). Meanwhile, the ability to rotate is modulated by the surrounding interommatidial cells: if these have high cortical tension and low neighbor exchange then rotation is inhibited, whereas if they have low cortical tension and high neighbor exchange then rotation is enhanced (and goes too far). Furthermore, the kinase Nmo controls cortical tension and hence presumably neighbor exchange in the interommatidial cells.

However, regarding the mechanism of action of Nmo, I found the data puzzling. It appears that total E-cad levels don't change in Nmo loss-of-function and gain-of-function conditions, and FRAP shows that both conditions reduce the stable fraction of E-cad at junctions, so both conditions are expected to reduce cell adhesion.

We appreciate that our earlier descriptions were confusing, and we are grateful for the reviewer's comments in guiding our experiments and conclusions.

We now present a careful analysis of the steady state levels of junctional E-cad. This is shown in the new Figure 5 for all three genotypes with an increase in junctional E-cad levels in *nmo* LOF and a decrease in *nmo* GOF backgrounds (Fig. 5a-c). This was done in a tissue and at a developmental stage of homogenous cell populations that is commonly used for such comparisons, pupal wings at 22h APF (see also Iyer et al 2019 from the Eaton lab for example). Such an analysis does not only allow the careful comparison of junctional E-cad levels in the different *nmo* backgrounds (new Figure 5a-c and also Ext Data figure 7e-h), but it also lends itself to a direct comparison with other related work (e.g. Classen et al 2005 and Iyer et al 2019 by the Eaton lab). The wing tissue is more suited for such analyses as eye tissue behind the furrow is

too dynamic, as it is a temporal progression with cells adopting different fates and thus the steady state of E-cad there is difficult to compare.

Importantly, we have completely revised the data presented for the FRAP analyses along the same lines, and now compare all three genotypes accordingly. These added data are now shown in the revised/new Figure 5 (panels 5g-i; and also Ext. Data Fig. 7a-d). See also below.

Yet, *Nmo* loss-of-function and gain-of-function conditions have opposite effects on ommatidial rotation and on junctional tension. I'm not sure if I missed something, or the authors missed something. However, regardless of the interpretation of the FRAP, there doesn't seem to be any direct evidence suggesting the mechanism is partly or entirely dependent on E-cad mediated cell adhesion and not on effects on actomyosin and the cortical cytoskeleton.

I've made suggestions for how the authors could address some of these issues, however the fundamental question of roles of adhesion vs tension probably requires some investigation of effects of *Nmo* on the actomyosin cytoskeleton.

Again, we have redone the FRAP analyses and comparison of the steady state junctional E-cad levels, as suggested by the reviewer and we believe the results are now much clearer. We thank the reviewer for their very helpful and constructive suggestions. We measured the recycling dynamics of all three genotypes (control, *nmo* LOF, and *nmo* GOF) by quantifying the initial recovery rate (new main Fig. 5g-i), which is independent of the overall amount of protein (or the stable fraction). Because the steady state levels are different, it makes much more sense in comparing the dynamics of the process.

Following the suggestion of the reviewer, we have removed the 1 μ m lateral diffusion FRAP experiment because it did not add anything to our study (as the reviewer mentioned).

As mentioned above, we have also quantified the steady state levels of junctional E-cad (new Fig. 5a-c). There is a clear difference between the steady state levels, with *nmo* mutant tissue having increased levels, and *Nmo* GOF having decreased levels (as compared to wt). We have removed the antibody uptake experiment (as suggested).

We hope that all of our additional data, expanded discussion, and removal of confusing data make the manuscript much clearer to the reader.

Regarding to the reviewer's comment about cytoskeletal involvement, we refer to the response to reviewer 1 above. Please see there, thank you. Along these lines, we did look at Zipper localization/levels in *nmo* mutant clones (Ext Data Fig 4f) and saw no difference, which is consistent with work from the Wolff lab (Fiehler and Wolff) suggesting that *nmo* and actomyosin regulation act in different "pathway" in the rotation process. We have confirmed this genetically ourselves (not shown).

Minor points:

We thank the reviewer for these careful and detailed points.

First sentence of abstract - "the molecular basis of tissue fluidity remains unknown" seems a bit of a sweeping statement. In some cases a good case can be made for processes such as cell intercalation (as in ref. 1) underlying fluidity. So maybe this statement should be qualified in some way? (Or removed?)

We have rephrased this to 'molecular regulators' to emphasize we are looking at methods to developmentally regulate the degree of fluidity and not the specific cell biological basis of fluidity. We hope this is clearer.

Second sentence of abstract - "OR": I tripped over this (new?) abbreviation every time I read it. Would it increase readability to simply say "ommatidial rotation" instead? Hopefully space is not so limited that this would be an issue. (The same might be said of "MF" and "IC", at least for readers not familiar with eye development.)

We appreciate this concern regarding abbreviations (it was done within the length restrictions of a Nature letter format). We now use 'rotation' or "rotation process" in the text to make it easier for readers not familiar with the field. We think abbreviating interommatidial cells to ICs, does make it easier to read and understand the text.

Second sentence of abstract - how similar is ommatidial rotation to cell migration? I think ommatidial rotation occurs in an epithelium where cells retain adhesive junctions, while migrating cells undergo EMT thus significantly altering their properties. I suspect the authors may be trying to make their favorite model sound more broadly relevant, but I don't think this is necessary (and if it is, what about an example of cell intercalation in a vertebrate epithelium?).

Ommatidial rotation has been accepted as a PCP-regulated cell motility paradigm. While it is not cell migration in a classical sense, it shares many features with other PCP-regulated cell motility processes like convergent extension processes in vertebrate gastrulation and neurulation

Final sentence of abstract - "Our study defines Nemo as a molecular tool" is quite ambiguous. Do the authors mean a tool for investigators to use (i.e. the conventional sense) or do they mean "the tissue" uses Nemo as a "tool"? Is it easier to say "modulation of Nemo activity is a molecular mechanism"?

We apologize for this confusing statement. We have adjusted that in the abstract. We also provide a better explanation of Nmo as a molecular tool in the discussion.

p.3 - "it remains largely unknown how fluidity of a tissue is regulated at the molecular level": this statement might come as a shock to the many other labs studying cell intercalation in Drosophila (Zallen, Lecuit, Bellaiche, Eaton and many others). Regulation of E-cad and actomyosin come to mind as the major mechanisms (with I think a number of studies at stages where proliferation is absent or sufficiently low to not play a major role?).

We appreciate the reviewer pointing out this confusing statement. We meant how developmental signals can alter tissue mechanics and fluidity, but understand that this did not come across as we intended. We have rewritten this section of the introduction to make it clearer, and included a much increased number of references covering the topic.

p.5 - "yet cluster composition and rotation direction remain unaffected": can we be sure of this by just looking at photoreceptor markers in eye discs or in adult eye sections? I agree the clusters look "normal" but has anyone ever looked at e.g. cone cells or other cell types recruited subsequently?

Thank you for indicating this as being confusing. We have rephrased the text to avoid this confusion and to add clarity. We have also added all the respective references in support of the general tissue requirement of Nmo and the largely normal cellular appearance of *nmo* mutant cells.

p.6 - "Consistently with this data, inhibition of cell delamination did not affect OR": I agree the clusters achieve the correct degree of rotation based on the adult eye sections, but it's not obvious this demonstrates that the process of ommatidial rotation is completely normal (it could for instance be delayed and then catch up). The same applies to oriented cell divisions. So maybe these statements could be toned down?

We fully agree with the reviewer and have rewritten this accordingly.

Is SV2 ever cited in the text?

We thank the reviewer for pointing this out, and SV2 is now cited accordingly.

p.7 - "Taken together our data suggest that the ICs serve a permissive role during the OR process." I agree with the general conclusion based on the findings thus far. However, the rest of the manuscript goes on to show that different interommatidial cell behaviors result in different degrees of rotation, which seems like an instructive role?

We have rephrased the text to say that ICs do not instruct the direction of rotation. Based on our data that the force-generating mechanism and direction are driven by the cluster itself, we propose that the ICs

function as a permissive substrate allowing the correct degree (and speed) of rotation, and that too much or too little fluidity causes the cluster to be stopped prematurely or overshoot the 90 degree mark – thus it limits how well the force-generating machinery can affect rotation, but does not actively pull the cluster around to add to the force-generation/rotation driving force.

p.8/ Fig.2D and H - in Fig.2D the averaged values show Nmo GOF showing pretty similar average ommatidial area to wt, yet the example of an individual ommatidium in Fig.2H gives quite a different profile. Presumably to get the average profile seen in Fig.2D, some individual ommatidia would have to contract faster than wt, if others are contracting more slowly?

That is correct. There is more variability in the GOF phenotype, as evidenced by the SD cloud size. Importantly, in the *nmo* mutant the cluster constricts very steadily with little variation, see small SD cloud in Fig 3c, whereas in the Nmo GOF genotype the process is more erratic. This is largely due to the fact that the expansion phases of the constriction process are larger in Nmo GOF than wt and expansion is almost non-existent in the *nmo* mutant (see also additional single ommatidia examples in new Ext. Data Fig. 6. The statistical analyses reveal that there is no significant difference in the constriction phases (Fig. 3e), but the differences are highly significant in the expansion phases (Fig. 3f).

p.9 - “whereas *nmo* GOF revealed enhanced cell displacement”: doesn’t Fig.2L show no difference in cell displacement for Nmo GOF? I’m also not sure “velocity” is increased relative to wt as in Fig.S4B it seems to start rather lower than wt and then converge to a wt level. The only evidence for Nmo GOF boosting neighbor exchange seems to be Fig.2G (plus possibly the delamination data)?

Thank you for this correction – we have edited the text accordingly. Importantly, we have also added data in the new Figure 3k to better represent the increased/decreased cellular movement in the respective genotypes.

p.10 - “E-cad fluorescence recovery can be achieved either through lateral movement within the plasma membrane or by recycling from the vesicular/endosomal compartment” or (for completeness) by delivery of newly synthesized protein?

We have rewritten the text and included a reference to the processes involved in recovery.

p.10 - “While no difference was observed in the first experimental set up between the three genotypes (Extended Data Fig. 6e-h)”: actually only show wt and *nmo* LOF? Is *nmo* GOF the same? I’m also surprised the authors conclude wt and *nmo* LOF are the same. My understanding is that without an a priori power calculation, just looking at the p-value doesn’t tell you that the populations are not different.

We have removed this data and all references to it.

Fig.3 and Fig.S6 - would it be better to put both *nmo* LOF and GOF FRAP in the main figure (on the same graphs)? This would make comparison easier for the reader.

We thank the reviewer for this suggestion and have added new FRAP graphs with all genotypes in the new Figure 5 and Extended data figure 7.

p.10 - the authors’ interpretation of the FRAP data seems to depend on the assumption that all recovery is via recycling. Not only does this ignore the possibility of diffusion (which doesn’t seem to be ruled out as a mechanism) but also of new synthesis. It would seem simpler at this stage in the story to stick to the basic message that *nmo* LOF shows lower flux of mobile E-cad and *nmo* GOF shows higher flux of mobile E-cad without presupposing the exact mechanism? (Although I’m unsure this measure is actually a useful one, see next comment.)

We appreciate the reviewer’s thoughtful comments. We have added new data showing the steady state levels differ in each genotype, and so have removed some of the assumptions that we made in the original draft. We have focused only on the initial recovery rate, and the half recovery time, both of which indicate a slower process in *nmo*LOF and a faster process in *nmo*GOF. We have added in new genetic interaction data with factors that specifically affect the endocytic recovery, with dynamin/*shibire* LOF suppressing the *nmo*GOF rotation phenotype, and deletion the p120 binding site in E-cad (which should increase recycling)

enhancing the phenotype. This data is added to the new Figure 5 and Extended data figure 7.

p.10 (last words) - "The pupal wing..." actually should say "prepupal" as indicated in Methods? Also p.11 3rd line from bottom "The Drosophila pupal wing", should this also be prepupal wing? The text implies measurements of apical area and hexagonal packing (Fig.3G-H) are all in pupal wings but without indicating the age, however earlier it is suggested that Fig.3F,G are pulse-chase data in prepupal wings? (Actually in the preceding section it is often hard to know if the data being described is pulse-chase as it says things like "E-cad antibody at AJs exhibit a higher junctional intensity in nmo mutant cells, as compared to controls" which could be taken to mean steady state not pulse-chase measurements.) Reworking this section of the Results to be more explicit would be very helpful.

We thank the reviewer for their constructive feedback to improve our paper, and we have accordingly removed all pulse-chase data, added new steady state analyses and rewritten this section.

p.10-11 - I agree that the E-cad antibody uptake experiment is the right one to measure rates of E-cad internalization from junctions (albeit I'm not sure how much this tells us about strength of adhesion), but if this is relevant shouldn't this be done in eye discs? It is not obvious that Nmo function will be the same in wings. Such assays have been published in wing discs where apparently the peripodial membrane is sufficiently ruptured during dissection and I would think the same was certainly true at the back of the eye disc.

- Also, as stated in the text, to show the change is due to internalization you need to quantify levels that appear in cytoplasmic vesicles (preferably showing they are Rab5 positive) but the authors say that strong junctional labeling prevents this, but such vesicles would not be expected to overlap with the junctions in xy or z, which maybe indicates that they are actually not seeing significant internalization over the course of the experiment? Furthermore, to be sure there is actually any change in junctional levels, we need to see a timecourse (i.e. no internalization, 5 min, 10 min etc)

We thank the reviewer for their comment – we actually meant that staining of the junction was so strong it was saturating before being able to see vesicles, rather than physical overlapping of structures. However, we have removed this discussion of cytoplasmic vesicles because it was confusing to the reader and did not add anything.

We have added new analysis of steady state junctional E-cad levels in Fig 5a-c, and a new set of FRAP analyses in Fig. 5g-i. With the new data as presented, we believe that the comments have been addressed adequately and the paper has been improved accordingly.

- And what happens in nmo GOF, do you see the expected increase in flux of the mobile E-cad fraction? As mentioned above, we have added new data from FRAP analyses and junctional E-cad steady state levels to document better the different effects of NmoGOF and nmo mutant tissue on E-cad dynamic. The new data and presentation nicely document that Nmo GOF does behave in the opposite manner to nmo LOF.

- Perhaps the simplest thing for the authors would be to remove the current data from the manuscript? A better use of the space might be to show steady state E-cad levels (possibly in permeabilised vs unpermeabilised, but this is less important) in wt, LOF and GOF eye discs? (See major point below.)

We did remove some of the problematic data as discussed above. Due to technical limitations it was much easier and reliable to perform the analysis in the wing tissue, where the cells are uniform. As we have added a whole new figure comparing the fluidity parameters between the genotypes (new Figure 4) and see the same effect in both the wing and the eye, we trust that these new wing quantifications are adequate.

Although the authors mention it a few times (e.g. in the title or statements such as "Nmo levels directly affect AJs downstream of PCP") I found it hard to see the relationship to PCP, which seems to have a separate role in cell fate specification and determining direction of rotation, rather than directly regulating Nmo (e.g. spatial patterns of Nmo activity). Mirkovic et al 2011 seems to imply Vang/Pk localize Nmo activity (at the R3/R4 boundary?) but I don't see how this fits the authors' current model for the role Nmo in ommatidial rotation. Isn't this all about Nmo function outside the photoreceptor cluster, with no obvious role

for planar cell polarized Nmo activity? And isn't the same true in the wing? Otherwise you would expect planar polarized differences in junctional tension (which would be easy to measure). Or are Vang/Pk playing a non-PCP role in generally recruiting Nmo to membranes (in which case it isn't strictly speaking "downstream of PCP" but "downstream of PCP factors playing a non-PCP role"?).

We appreciate the reviewer's thoughtful comments. We have spent a good deal of time considering the phrasing and believe there is some value in describing Nmo as acting downstream of PCP for the following reasons:

Planar polarity in the adult eye is manifest as the alignment of ommatidia within the D/V axis, similarly to the alignment of wing hairs, although there are many more complicated steps involved. In addition to affecting R3/R4 fate and rotation direction, several PCP mutants also have 'degree of rotation' phenotypes, e.g. *Vang* and *pk* null phenotypes. Nmo is thus affecting the final planar polarized output at the tissue level, as well as genetically acting in a pathway with PCP proteins acting in a planar polarity context.

As cells are initially recruited into the pre-cluster (just after the morphogenetic furrow passes), all cells are polarized in the D/V axis responding to the Wg/dWnt4 gradient at each pole, subsequently the PCP proteins are upregulated in the cluster, but they are expressed at low levels, and are localized to the D/V border in each cell. Within this study we have not looked at endogenous Nmo localization but we would not be surprised if it were initially polarized across the tissues (including ICs) based on the activity of PCP complexes.

Based on published work from the Eaton and Strutt labs, there does appear to be a planar polarized increase in tension in the proximo-distal axis of the pupal wing (Iyer et al 2019) and the E-cad distribution at 'vertical' vs 'horizontal' junctions in the pupal wing is regulated by PCP factors (both in steady state and FRAP experiments – Warrington et al 2012). Moreover, the cell packing in the wing and its visco-elastic properties (closely related to tissue fluidity) are both heavily dependent upon E-cad recycling and PCP factors (Iyer et al 2019 and Classen et al 2005). The link between PCP/E-cad recycling/tissue fluidity also extends to other planar polarized movements such as convergent extension and axis elongation in flies and vertebrates (see references in main paper text). We have expanded our discussion of these processes, and how Nmo might be involved in PCP processes more generally.

Thus, although the molecular interactions of PCP factors and Nmo are outside the experimental scope of this manuscript, we believe the published literature supports the discussion of Nmo acting downstream of PCP factors, as was shown in Mirkovic et al 2011. We thank the reviewer for having challenged our hypothesis and making us consider this question in more detail. We trust that the new data we have added, and our elaborated discussion tie in our work to the published studies on PCP and E-cad trafficking to a satisfactory extent and increase the impact of our work.

Major points:

p.10 - There is an anomaly in the FRAP data which seems hard to explain, which is that both *nmo* LOF and GOF result in higher mobile fractions of E-cad at junctions. If total E-cad levels remain constant at junctions in *nmo* LOF and GOF (which I think these authors published previously?) then both *nmo* LOF and GOF will have less immobile (i.e. bound) E-cad at junctions and thus will be expected to have weaker adhesion. This seems to be at odds with the model of the authors that *nmo* LOF has more stable junctions with fewer neighbor exchanges while *nmo* GOF has less stable junctions with more neighbor exchanges. I don't think the rate of turnover of the mobile fraction (which the authors seem to focus on) is the relevant measure here, except in as much as it might contribute to the size of the immobile fraction (a relationship that seems murky) and thus inferred strength of adhesion.

We thank the reviewer for drawing our attention to this assumption and have performed additional experiments in Fig 5a-c (as outlined above), showing that steady state levels of junctional E-cad are indeed different with opposite effects of *nmo* LOF and GOF. Similarly, the new FRAP data analyses are consistent with the notion that junctional E-cad is less dynamic in *nmo* LOF and more dynamic in *nmo* GOF, and accordingly affects tissue fluidity. We appreciate that our previous presentation and interpretation was confusing (apologies for that), and we have adjusted the results and discussion sections with the new data

sets (as mentioned above throughout) accordingly.

p.11 - "This confirms our interpretation of the FRAP experiments that the higher recovery level post-bleach would result in increased E-cad steady state levels at AJs." (See also section heading on p.9 "Nmo regulates E-cad levels at AJs".) I'm not sure if I missed something, but aren't steady state levels the same in wt, nmo LOF and nmo GOF? On p.5 it is stated "nmo does not affect the overall E-cad levels" and I don't recall seeing new data in the manuscript to contradict this. Changes in steady-state levels would help to explain some of the puzzling aspects of the data, so maybe this should be re-investigated?

We thank the reviewer for this suggestion. It has considerably improved our manuscript.

p.12 - the laser ablation experiments give a pleasingly clear result, showing increased junctional tension in nmo LOF and decreased in nmo GOF. However, I don't see why the authors think this shows this is (entirely) due to changes in E-cad mediated adhesion. On p.13 the authors say "our data indicate that the control of junctional E-cad levels and dynamics can directly modulate tissue tension, viscoelasticity, and stress, thus affecting tissue fluidity and cell motility" but this seems at best speculation. As noted above, nmo LOF and GOF seem to both reduced levels of immobile E-cad at junctions, so both would be expected to have decreased adhesion. If this is true, it seems rather more likely that Nmo is acting by modulating actomyosin and thus cortical tension directly, with LOF and GOF having opposite effects.

Again, we apologize for our earlier confusion. We have changed the text in the results and conclusions and added to the discussion of E-cad dynamics and tissue tension, and associated tissue fluidity and jamming/unjamming transition as regulated by Nmo.

REVIEWERS' COMMENTS

Reviewer #1 (Remarks to the Author):

A solid revision. I would recommend publication.

Reviewer #2 (Remarks to the Author):

The authors have largely addressed my concerns. The additional studies in the wing and the cell shape analyses have strengthened the authors' conclusion that IC tissue fluidity plays a permissive role in ommatidial rotation. The changes to the manuscript, extended discussion, and additional literature references have strengthened the manuscript. A remaining weakness is that Ecad levels and dynamics are measured in the wing but not in the eye, but this appears to be technical limitation of the eye system. The addition of the cell shape analyses and associated predicted tissue mechanical properties are a very nice addition that support their interpretations. I have one question related to this new data: it appears that the elongated cell shapes in the nmo GOF cells are all aligned along the same direction (Fig. 4F). What explains this? Would one expect a more isotropic distribution in a fluid-like tissue?

Reviewer #3 (Remarks to the Author):

As stated e.g. at the beginning of the Discussion, the authors have two main conclusions. The first “This work defines a novel role for Nmo kinase in modulating tissue fluidity” is well-supported and the additional analysis in Fig.4 on “jamming” further supports this conclusion. The second take-home message that “Nmo regulates the dynamics of E-cad turnover at AJs directly, thus impacting junctional remodeling and tissue fluidity” still feels rather shaky.

When the authors say Nmo acts “directly” to regulate E-cad dynamics, I assume they are referring to the data in Mirkovic et al (ref.33) showing Nmo can phosphorylate E-cad and b-catenin, however that study (or this study) doesn't establish a role of those phosphorylation events in E-cad dynamics – thus this assertion feels like an over-stretch. Similarly, even if Nmo is affecting E-cad dynamics (a perfectly reasonable idea), the presented data are rather unclear on whether the changes in E-cad reported are actually consistent with the under- and over-rotation phenotypes of Nmo – an issue that was also

evident in the previous version of the manuscript (although new data is moving in an encouraging direction).

In terms of the question of whether Nmo acts primarily (or directly) via modulation of E-cad or instead some other adhesive or cytoskeletal components, one additional argument used by the authors is based on genetic interactions. However, the interaction of *sec>Nmo* with *shi/+* simply suggests that endocytosis is involved at some level in ommatidial rotation, but doesn't really point the finger at E-cad. Similarly, the interaction with *mys/+* if anything points the finger away from E-cad. The UAS-*shg* and UAS-*shgdJM* is more encouraging but doesn't prove Nmo normally acts through E-cad but only that changing E-cad can modulate rotation (and these seem to be random P-element inserts, so I'm not sure how we know they are expressed at the same level). On the whole I felt the genetic interaction data undermined the story (and if I were the authors I might take it out?). A better way of approaching this might be to modulate E-cad levels up and down (e.g. E-cad^{-/-} vs addition of a copy of e.g. *ubi-E-cad-GFP*) at the same time as modulating *nmo* (mut and GOF) and look for changes in E-cad stable amount at junctions, alongside a comparison to degree of rotation. If E-cad stable amount consistently correlated with rotation phenotype, and Nmo had a consistent effect on E-cad, then I think this would nail the story.

Regarding the question of whether Nmo affects E-cad in a way consistent with its effects on rotation, the central issue seems to be modulation of degree of E-cad-mediated cell adhesion. It seems intuitively obvious that more E-cad in stable adhesion will equal more adhesion (and less rotation) and less E-cad in stable adhesions will give weaker adhesion (and more rotation) – and I think this is what the authors are arguing. In this regard, they have a very encouraging result, in that at least in the pupal wing *nmo*[mut] causes an increase in E-cad at junctions and *nmo*[GOF] causes a decrease (although this isn't verified in the eye, and stands in contrast to previous reports that E-cad junctional levels don't change in *nmo*[mut] in the eye).

However, even for the pupal wing, this analysis of E-cad levels is not extended to asking how much of this E-cad is stable (e.g. by the criterion of FRAP) and thus likely to be contributing to adhesion. The authors instead now concentrate on the rate of recovery of E-cad in junctions of ICs, and seem to conclude that faster recovery in *nmo*[GOF] implies lower adhesion and slower recovery implies higher adhesion – but this is simply describing behavior of the mobile fraction which I'm not sure is even expected to contribute to adhesion? I can see there might be a correlation between faster turnover and a smaller stable amount, but I think this would have to be demonstrated for any particular case.

The authors do still include the stable fraction data (Ext. Data 7d) but don't seem to discuss it any more. Although it is mixing eye and wing data, I think from this you can try to infer stable amounts of E-cad in adhesions. Compared to wt, *nmo*[GOF] seems to have 0.75x less total E-cad at junctions (in the wing) and *nmo*[mut] has 1.5x more, and a *nmo*[GOF] has ~45% stable fraction and *nmo*[mut] ~55% vs ~70% in wt (in the eye) – so that implies 0.7 units of stable E-cad in wt, 0.34 stable E-cad in *nmo*[GOF] and 0.8 in

nmo[mut]? These numbers seem to make sense and support the authors' conclusions, however, they really need confirming in a single tissue (mixing between eye and wing isn't really appropriate). I also have some concerns about the statistics used on the E-cad data (and elsewhere), as the numbers of biological replicates (i.e. the "n" number) seems low (and incorrectly assumed to be the number of sample points within an individual and not between individuals).

Major points:

Lines 351-353 – I understand the authors' argument for not attempting E-cad quantitation in the eye disc, but I feel it would be important to see a qualitative difference e.g. inside and outside loss- and gain-of-function clones in the eye. From the pupal wing images it seems as if the differences are large, so this would presumably also be evident in the eye? (And a large difference probably could be quantified.)

Lines 375-379 describes the eye disc FRAP analysis, but unlike the previous version of the manuscript now only considers the rate of recovery and not the size of the stable fraction (the recovery fraction is actually shown in Extended Data Fig.7d but never discussed in the text). I'm unsure whether this is actually the same data set as presented previously, but my main concern regarding the previous data was that both nmo[mut] and nmo[GOF] showed a smaller stable fraction of E-cad, so (on the face of it) might both increase tissue fluidity. The authors now only discuss rate of turnover of the unstable fraction, but I assume as this fraction is unstable it probably doesn't contribute much to cell adhesion and thus tissue fluidity. The issue of why both nmo[mut] and nom[GOF] decrease the stable fraction of E-cad but have opposite phenotypes may well be resolved by the observation that steady state levels are in fact different in these genotypes. However, to firm up this argument, the authors really need to show (in the same tissue, ideally the eye) that relative stable amount of E-cad at junctions is higher in nmo[mut] and lower in nmo[GOF]. As the pupal wing seems more amenable to quantitation, it seems like the authors ideally should do FRAP on pupal wings to determine the stable fractions of E-cad, then use these to calculate a stable amount based on the steady state levels (which would be revealed by the E-cad-GFP fluorescence levels prior to bleaching?) for wt vs nmo[mut] and nmo[GOF].

Line 1054 – the very low p values appear to be the result of assuming n=300. However, I think the correct biological replicate is a wing from another animal (see e.g. doi.org/10.1371/journal.pbio.2005282 or doi.org/10.15252/embj.201592958), not a junction from the same wing (which is a technical replicate), so surely n=3? Are the results still significant with n=3 (I would think so?)?

Line 1061 – again the p value of 10^{-6} seems very low. It looks a bit like FRAP may have been done on only 1 eye disc (line 1072), so shouldn't n=1? In which case can stats be done, as any one sample could just be an "abnormal" disc? p-value for panel j is also concerning, but I can't spot where the n numbers are for this experiment.

Looking carefully at all the figures, I realize that this failure to correctly distinguish between technical (intra-individual) and biological (inter-individual) replicates seems to affect all the stats (even though the results might well be significant if correctly analyzed). For instance Fig.4g reports $p=7.64 \times 10^{-62}$ and 4.49×10^{-141} – these very low p-values are rather implausible when n=2 (number of pupae).

Minor points:

Line 33 “the molecular regulators of tissue fluidity remain unknown” – I still wonder if someone might take issue with the idea that no molecular regulators are known? Maybe “poorly characterized”, “largely unexplored”? I note line 80 is more nuanced and says “remain largely unknown”.

Lines 33-34 “driven by planar cell polarity signaling” – but I think it still happens in absence of planar cell polarity signaling? So is it “driven” or maybe instead “directed” (i.e. the direction)? (Same comment also applies to Title) See also lines 128-130 which talk about “PCP-directed” which seems better.

Lines 295-348 form one long paragraph, but it might be easier to read if broken up a bit into shorter paragraphs (particularly as this is very nice new data).

Lines 338-339 “In each case, the degree of tissue fluidity impacts the ability of the cluster to rotate the full 90°” – I agree the correlation is striking and the conclusion probably correct, but it seems perhaps premature to say one impacts the other at this point in the manuscript, before a possible mechanism is established?

Lines 324-330 and 341-346 – both sections of text deal with reduction in hexagonal packing and I think are saying the same thing, albeit with an emphasis on E-cad recycling in one case and tension in the other (interfacial tension caused by changes in E-cad?). The repetition is a bit confusing and I wonder if the sections could be combined?

Line 347 “E-cad recycling rates” – really turnover as measured by FRAP, so could be diffusion or new synthesis as well as recycling.

Lines 360-363 “These data are consistent with reduced E-cad endocytosis leading to higher levels at the junctions in nmomut LOF cells, and increased endocytosis leading to lower levels of junctional Ecad in the GOF cells” – this could be true, but so far no data has been presented on endocytosis (it could for instance be a difference in levels of transcription or recycling) so it seems strange to jump the gun with this conclusion. (“E-cad” has also lost its hyphen here.)

Lines 373 – would be helpful if the text here indicated that FRAP was carried out in junctions between ICs. This is evident in the figure, but the previous paragraph in the text was about pupal wings.

Extended Date 7g – I wasn’t sure I could see where this was cited in the text.

REVIEWERS' COMMENTS

Reviewer #1 (Remarks to the Author):

A solid revision. I would recommend publication.

We thank the reviewer for their positive appreciation of all our hard work that went into the revision of the paper. It is a rare situation these days, which we savor and truly appreciate. Thank you very much!

Reviewer #2 (Remarks to the Author):

The authors have largely addressed my concerns. The additional studies in the wing and the cell shape analyses have strengthened the authors' conclusion that IC tissue fluidity plays a permissive role in ommatidial rotation. The changes to the manuscript, extended discussion, and additional literature references have strengthened the manuscript. A remaining weakness is that Ecad levels and dynamics are measured in the wing but not in the eye, but this appears to be technical limitation of the eye system. The addition of the cell shape analyses and associated predicted tissue mechanical properties are a very nice addition that support their interpretations. I have one question related to this new data: it appears that the elongated cell shapes in the nmo GOF cells are all aligned along the same direction (Fig. 4F). What explains this? Would one expect a more isotropic distribution in a fluid-like tissue?

We appreciate this observation, thank you.

To clarify this point:

Anisotropic tension in the proximal-distal/P-D axis is due to adhesion via Dumpy and hinge contraction as nicely described previously (Aigouy et al, Cell 2010; Etournay et al, Elife 2015, PMID 26102528).

In the context of our work, it is worth noting that the altered property of E-cad trafficking in p120 mutant wings results in increased cumulative shear stress, from the same anisotropic P-D axis tension, leading to cell elongation in the P-D axis (Iyer et al, Curr Biol. 2019). We observe the very same effect(s) with the nmo LOF and GOF scenarios, which serves as one of the points to indicate that Nmo affects junctional stability via E-cad. We trust that with the added discussion and comparison to the data presented by Iyer et al (2019) this is adequately covered in the revised paper.

Reviewer #3 (Remarks to the Author):

As stated e.g. at the beginning of the Discussion, the authors have two main conclusions. The first "This work defines a novel role for Nmo kinase in modulating tissue fluidity" is well-supported and the additional analysis in Fig.4 on "jamming" further supports this conclusion. The second take-home message that "Nmo regulates the dynamics of E-cad turnover at AJs directly, thus impacting junctional remodeling and tissue fluidity" still feels rather shaky.

When the authors say Nmo acts "directly" to regulate E-cad dynamics, I assume they are referring to the data in Mirkovic et al (ref.33) showing Nmo can phosphorylate E-cad and β -catenin, however that study (or this study) doesn't establish a role of those phosphorylation events in E-cad dynamics – thus this assertion feels like an over-stretch. Similarly, even if Nmo is affecting E-cad dynamics (a perfectly reasonable idea), the presented data are rather unclear on whether the changes in E-cad reported are actually consistent with the under- and over-rotation phenotypes of Nmo – an issue that was also evident in the previous version of the manuscript (although new data is moving in an encouraging direction).

In terms of the question of whether Nmo acts primarily (or directly) via modulation of E-cad or instead some other adhesive or cytoskeletal components, one additional argument used by the authors is based on genetic interactions. However, the interaction of *sec>Nmo* with *shi/+* simply suggests that endocytosis is involved at some level in ommatidial rotation, but doesn't really point the finger at E-cad. Similarly, the interaction with *mys/+* if anything points the finger away from E-cad. The *UAS-shg* and *UAS-shgdJM* is more encouraging but doesn't prove Nmo normally acts through E-cad but only that changing E-cad can modulate rotation (and these seem to be random P-element inserts, so I'm not sure how we know they are expressed at the same level). On the whole I felt the genetic interaction data undermined the story (and if I were the authors I might take it out?). A better way of approaching this might be to modulate E-cad levels up and down (e.g. *E-cad/-* vs addition of a copy of e.g. *ubi-E-cad-GFP*) at the same time as modulating *nmo* (mut and GOF) and look for changes in E-cad stable amount at junctions, alongside a comparison to degree of rotation. If E-cad stable amount consistently correlated with rotation phenotype, and Nmo had a consistent effect on E-cad, then I think this would nail the story.

We responded to all the reviewers' comments and queries to the original submission that suggested *nemo* might be acting on the cytoskeleton rather than through E-cad trafficking. All the new data focus on further supporting the model that Nemo acts via E-cad junctional dynamics.

In brief: we have used validated and available tools to tackle the question of E-cad trafficking in OR in multiple ways – *shi/+*, which affects endocytic recycling of E-cad (and other proteins) and in similar contexts has been shown to affect E-cad mediated cell rearrangements and shape change (Classen et al, Dev Cell 2005, as a classical example). We have focused on examining heterozygous animals and so processes that are heavily dependent upon Shi activity, such as E-cad recycling during morphogenesis and junctional remodeling, will be most severely affected rather than a complete block on all dynamin function (which would not be informative). We demonstrate that *shi/+* can suppress the Nmo GOF phenotype, suggesting that the Nmo GOF phenotype is in part due to dynamin-dependent endocytosis. We also used well described, validated, and molecularly understood *UAS-Shg(E-cad)* tools (from the lab of Pernille Rorth, where they were used in border cell migration) to further show that deleting the p120 binding site, which would promote E-cad trafficking, enhances the GOF ommatidial rotation phenotype. All these data are fully consistent with the notion that Nmo affects E-cad trafficking/recycling and is in full support of our model and conclusions. Generating new tools and transgenic constructs/flyes was outside of the scope of this work.

Also, we used the *mys/+* background to show that the *Sev>Nmo* phenotype is sensitive to known adhesion-modulators of ommatidial rotation, and also in response to (another) reviewer's comments, who suggested including more discussion of Integrin-mediated adhesion in ommatidial rotation. The experiments were designed to ask and confirm that Nmo phenotypes are related to E-cad trafficking rather than downstream cytoskeletal effects.

The reviewer is now suggesting additional experiments at this revised manuscript stage that were not suggested in the initial review, which is not appropriate.

In addition, we have performed experiments or altered the text to address each point raised by each reviewer in the initial review to address, clarify, and confirm the points raised. All our data are consistent with the notion that Nemo affects E-cad trafficking/recycling, and we trust that the data as a whole are very convincing and well integrated with related work in different tissues.

Regarding the question of whether Nmo affects E-cad in a way consistent with its effects on rotation, the central issue seems to be modulation of degree of E-cad-mediated cell adhesion. It seems intuitively obvious that more E-cad in stable adhesion will equal more adhesion (and less rotation) and less E-cad in stable adhesions will give weaker adhesion (and more rotation) – and I think this is what the authors are arguing. In this regard, they

have a very encouraging result, in that at least in the pupal wing *nmo[mut]* causes an increase in E-cad at junctions and *nmo[GOF]* causes a decrease (although this isn't verified in the eye, and stands in contrast to previous reports that E-cad junctional levels don't change in *nmo[mut]* in the eye).

This comment supports what we are saying. The reviewer has failed to note that we have concluded that junctional levels do change in the eye – the previous statements were unsupported.

However, even for the pupal wing, this analysis of E-cad levels is not extended to asking how much of this E-cad is stable (e.g. by the criterion of FRAP) and thus likely to be contributing to adhesion. The authors instead now concentrate on the rate of recovery of E-cad in junctions of ICs, and seem to conclude that faster recovery in *nmo[GOF]* implies lower adhesion and slower recovery implies higher adhesion – but this is simply describing behavior of the mobile fraction which I'm not sure is even expected to contribute to adhesion? I can see there might be a correlation between faster turnover and a smaller stable amount, but I think this would have to be demonstrated for any particular case. We trust that addition of all the new data addresses the points in question. With all due respect to the reviewer's suggestion, it is simply not possible to do what the reviewer is requesting, to perform FRAP experiments "for any particular case". We trust that the addition of the new data of the highly informative *shape factor*-analyses in two tissues, eye and wing, and its correlation with increased or reduced adhesion behavior and tissue stiffness/jamming addresses this point elegantly.

The authors do still include the stable fraction data (Ext. Data 7d) but don't seem to discuss it any more. Although it is mixing eye and wing data, I think from this you can try to infer stable amounts of E-cad in adhesions. Compared to wt, *nmo[GOF]* seems to have 0.75x less total E-cad at junctions (in the wing) and *nmo[mut]* has 1.5x more, and a *nmo[GOF]* has ~45% stable fraction and *nmo[mut]* ~55% vs ~70% in wt (in the eye) – so that implies 0.7 units of stable E-cad in wt, 0.34 stable E-cad in *nmo[GOF]* and 0.8 in *nmo[mut]*? These numbers seem to make sense and support the authors' conclusions, however, they really need confirming in a single tissue (mixing between eye and wing isn't really appropriate). We thank the reviewer for acknowledging in this comment that our data and interpretation "make sense".

We show conclusively by the *shape factor*-analyses and comparisons in two tissues, eyes and wings, and between tissues that Nemo has the same effects on adhesion and tissue fluidity/jamming in both contexts. It is thus valid and appropriate that we can use conclusions from both tissues in support of the same underlying model.

I also have some concerns about the statistics used on the E-cad data (and elsewhere), as the numbers of biological replicates (i.e. the "n" number) seems low (and incorrectly assumed to be the number of sample points within an individual and not between individuals).

With all due respect, we have to say that the reviewer is wrong. The *n* number is the number of individuals – either ommatidia or single cells, depending on the analysis. Each ommatidial cluster is independent from its neighbors in its phenotypic decision making in the PCP and ommatidial rotation context, as such the *n* value is the number of ommatidia analyzed in that context and not the number of animals. There is a very large body of research and references to document this, going back to elegant work in the Carthew lab in the mid 1990s (Zheng et al, Development 1995) and many studies since then. It is not the average across animals that matters but the distribution of phenotypic values between individual ommatidia. To document this further we have added the new Suppl. Figure S6, which demonstrates nicely that the behavior of individual animals is very similar within the same genotype from animal to animal (as expected), but very different between distinct genotypes.

Similarly, when analyzing cellular behavior and cell shape, individual cells are the n value, as was described and used many times in the elegant and groundbreaking work of the Eaton and Julicher labs, many of these references are also cited in our paper.

Major points:

Lines 351-353 – I understand the authors' argument for not attempting E-cad quantitation in the eye disc, but I feel it would be important to see a qualitative difference e.g. inside and outside loss- and gain-of-function clones in the eye. From the pupal wing images it seems as if the differences are large, so this would presumably also be evident in the eye? (And a large difference probably could be quantified.)

We appreciate the comment. We have performed the quantification in the tissue in which it was feasible. We have tried to perform similar analysis in the eye disc, but as the junctions are much smaller (shorter) and the processes are more dynamic, it is extremely difficult (to not say impossible) to quantify this in a static fixed eye disc image. That is the reason why we have added the pupal wing data, and confirmed that Nmo has similar effects on cell shape and tension, via the *shape factor*, and tissue fluidity in the wing as in the eye.

Lines 375-379 describes the eye disc FRAP analysis, but unlike the previous version of the manuscript now only considers the rate of recovery and not the size of the stable fraction (the recovery fraction is actually shown in Extended Data Fig.7d but never discussed in the text). I'm unsure whether this is actually the same data set as presented previously, but my main concern regarding the previous data was that both nmo[mut] and nmo[GOF] showed a smaller stable fraction of E-cad, so (on the face of it) might both increase tissue fluidity. The authors now only discuss rate of turnover of the unstable fraction, but I assume as this fraction is unstable it probably doesn't contribute much to cell adhesion and thus tissue fluidity. The issue of why both nmo[mut] and nmo[GOF] decrease the stable fraction of E-cad but have opposite phenotypes may well be resolved by the observation that steady state levels are in fact different in these genotypes. However, to firm up this argument, the authors really need to show (in the same tissue, ideally the eye) that relative stable amount of E-cad at junctions is higher in nmo[mut] and lower in nmo[GOF]. As the pupal wing seems more amenable to quantitation, it seems like the authors ideally should do FRAP on pupal wings to determine the stable fractions of E-cad, then use these to calculate a stable amount based on the steady state levels (which would be revealed by the E-cad-GFP fluorescence levels prior to bleaching?) for wt vs nmo[mut] and nmo[GOF].

We have focused on the rate of recovery, because this is (by far) the most relevant information in the context of junctional stability and dynamics. It also allows a much better comparison across different backgrounds, as the rate of recovery does not depend on initial levels, which are clearly different in the distinct genetic backgrounds, as addressed above and below. We added an explanation to the Methods section on why these are the best parameters to analyze. Thus, to make the FRAP analyses more generally approachable to a broad readership, the focus on the rate of recovery was the best way forward. It truly addresses the level of E-cad dynamics in junctions. We have added these data (which were new for the GOF background) to the main text. We have also added the specific quantifications of "*recovery rate*" and "*half-recovery time*" as informative box plots to the revised Figure 5.

In the revised paper, we have also added a thorough quantification of E-cad levels at the junctions (in pupal wings) in both LOF and GOF backgrounds and compared this to wild-type (as now requested by the reviewer... it was there in the revised submitted version). This nicely documents the significant differences of steady state junctional levels of E-cad in the LOF vs wt vs GOF backgrounds (one of the reviewer's suggested that this is as informative as the FRAP). These differences in levels correlate very well with the (cell) *shape factor* values and thus allow us to conclude that less or more E-cad is being turned over in the respective backgrounds. All these data are consistent with our conclusions and model.

To further corroborate this, we have also added *in vivo* genetic studies with *nmo-GOF* and *shibire/dynamin* to the revised Figure 5, which again confirms our model and is fully consistent with our conclusions. We trust that the new presentation of the FRAP data and the presence of all additional new data to the revised paper has addressed the comments to the original paper version, as was recognized by the other reviewers.

Line 1054 – the very low *p* values appear to be the result of assuming *n*=300. However, I think the correct biological replicate is a wing from another animal (see e.g. doi.org/10.1371/journal.pbio.2005282 or doi.org/10.15252/emj.201592958), not a junction from the same wing (which is a technical replicate), so surely *n*=3? Are the results still significant with *n*=3 (I would think so)?

We have consulted with several experts and we trust that our statistical analyses are correct. We thus respectfully disagree with the reviewer. Our analyses are fully consistent with many previous papers, most notably all from the collaborative effort of the Eaton and Julicher labs, who have set the high standard for such work.

In experiments like described in our paper, or those by the Eaton/Julicher labs, individual cells or ommatidia within the same animal are not technical replicates but independent units, as they behave independently of their neighbors. For example, the chirality of an ommatidium or its degree of rotation behaves fully independently of its neighbors, if the genetic control mechanisms are perturbed – see core PCP mutant phenotypes for instance. The same applies to junctions between cells.

The statistical analyses compare the distribution of values, not the averages of a given animal (which is a technical replicate), and as such these distributions - as they are rather different – represent very low *p* values being highly significant. To further support and explain this, we have added a new Supp. Figure S6 that documents that the distribution values for ommatidial rotation and constriction area between animals of the same genotype are highly similar, and also that these distributions are very different between the distinct genotypes.

Line 1061 – again the *p* value of 10^{-6} seems very low. It looks a bit like FRAP may have been done on only 1 eye disc (line 1072), so shouldn't *n*=1? In which case can stats be done, as any one sample could just be an “abnormal” disc? *p*-value for panel *j* is also concerning, but I can't spot where the *n* numbers are for this experiment.

Looking carefully at all the figures, I realize that this failure to correctly distinguish between technical (intra-individual) and biological (inter-individual) replicates seems to affect all the stats (even though the results might well be significant if correctly analyzed). For instance Fig.4g reports $p=7.64 \times 10^{-62}$ and 4.49×10^{-141} – these very low *p*-values are rather implausible when *n*=2 (number of pupae).

Again, we reiterate the statement for the point above. In experiments like reported in this study, individual cells or ommatidia within the same animal are not technical replicates. They are independent biological units, acting independently of their neighbors. Again, the chirality or degree of rotation of one cluster is independent of its neighbors – see PCP mutant phenotypes for instance. The same applies to cellular junctions. See point above for more detail.

Minor points:

Line 33 “the molecular regulators of tissue fluidity remain unknown” – I still wonder if someone might take issue with the idea that no molecular regulators are known? Maybe “poorly characterized”, “largely unexplored”? I note line 80 is more nuanced and says “remain largely unknown”.

Thank you, this has been adjusted.

Lines 33-34 “driven by planar cell polarity signaling” – but I think it still happens in absence

of planar cell polarity signaling? So is it “driven” or maybe instead “directed” (i.e. the direction)? (Same comment also applies to Title) See also lines 128-130 which talk about “PCP-directed” which seems better.

This has been corrected, thank you.

Lines 295-348 form one long paragraph, but it might be easier to read if broken up a bit into shorter paragraphs (particularly as this is very nice new data).

This has been adjusted accordingly. Thank you for appreciating this data as “nice”.

Lines 338-339 “In each case, the degree of tissue fluidity impacts the ability of the cluster to rotate the full 90°” – I agree the correlation is striking and the conclusion probably correct, but it seems perhaps premature to say one impacts the other at this point in the manuscript, before a possible mechanism is established?

This has been adjusted accordingly.

Lines 324-330 and 341-346 – both sections of text deal with reduction in hexagonal packing and I think are saying the same thing, albeit with an emphasis on E-cad recycling in one case and tension in the other (interfacial tension caused by changes in E-cad?). The repetition is a bit confusing and I wonder if the sections could be combined?

We appreciate the comment, but we trust that for clarity it is best to leave the text as is.

Line 347 “E-cad recycling rates” – really turnover as measured by FRAP, so could be diffusion or new synthesis as well as recycling.

This has been adjusted accordingly.

Lines 360-363 “These data are consistent with reduced E-cad endocytosis leading to higher levels at the junctions in nmomut LOF cells, and increased endocytosis leading to lower levels of junctional Ecad in the GOF cells” – this could be true, but so far no data has been presented on endocytosis (it could for instance be a difference in levels of transcription or recycling) so it seems strange to jump the gun with this conclusion. (“E-cad” has also lost its hyphen here.)

We trust the conclusion is well justified, we have adjusted the missing hyphen.

Lines 373 – would be helpful if the text here indicated that FRAP was carried out in junctions between ICs. This is evident in the figure, but the previous paragraph in the text was about pupal wings.

This has been added, thank you.

Extended Date 7g – I wasn’t sure I could see where this was cited in the text.

We apologize for not specifically referring to this panel. This has been corrected.